# Whole-genome sequencing and gene sharing network analysis powered by machine learning identifies antibiotic resistance sharing between animals, humans and environment in livestock farming

Zixin Peng[1‡], Alexandre Maciel-Guerra[2‡], Michelle Baker[2‡], Xibin Zhang[3], Yue Hu[2], Wei Wang[1], Jia Rong[3], Jing Zhang[1], Ning Xue[2], Paul Barrow[2,4], David Renney[5], Dov Stekel[6], Paul Williams[7], Longhai Liu[3], Junshi Chen[1], Fengqin Li[1]*, Tania Dottorini[2]*

1 NHC Key Laboratory of Food Safety Risk Assessment, Chinese Academy of Medical Science Research Unit (2019RU014), China National Center for Food Safety Risk Assessment, Beijing, People's Republic of China, 2 School of Veterinary Medicine and Science, University of Nottingham, Sutton Bonington, United Kingdom, 3 Qingdao Tian run Food Co., Ltd, New Hope, Beijing, People's Republic of China, 4 School of Veterinary Medicine, University of Surrey, Guildford, Surrey, United Kingdom, 5 Nimrod Veterinary Products Limited, Moreton-in-Marsh, United Kingdom, 6 School of Biosciences, University of Nottingham, Sutton Bonington, United Kingdom, 7 Biodiscovery Institute and School of Life Sciences, University of Nottingham, Nottingham, United Kingdom

‡ These authors are co-first authors on this work.
* lifengqin@cfsa.net.cn (FL); tania.dottorini@nottingham.ac.uk (TD)

## Abstract

Anthropogenic environments such as those created by intensive farming of livestock, have been proposed to provide ideal selection pressure for the emergence of antimicrobial-resistant *Escherichia coli* bacteria and antimicrobial resistance genes (ARGs) and spread to humans. Here, we performed a longitudinal study in a large-scale commercial poultry farm in China, collecting *E. coli* isolates from both farm and slaughterhouse; targeting animals, carcasses, workers and their households and environment. By using whole-genome phylogenetic analysis and network analysis based on single nucleotide polymorphisms (SNPs), we found highly interrelated non-pathogenic and pathogenic *E. coli* strains with phylogenetic intermixing, and a high prevalence of shared multidrug resistance profiles amongst livestock, human and environment. Through an original data processing pipeline which bcombines omics, machine learning, gene sharing network and mobile genetic elements analysis, we investigated the resistance to 26 different antimicrobials and identified 361 genes associated to antimicrobial resistance (AMR) phenotypes; 58 of these were known AMR-associated genes and 35 were associated to multidrug resistance. We uncovered an extensive network of genes, correlated to AMR phenotypes, shared among livestock, humans, farm and slaughterhouse environments. We also found several human, livestock and environmental isolates sharing closely related mobile genetic elements carrying ARGs across host species and environments. In a scenario where no consensus exists on how antibiotic use in the livestock may affect antibiotic resistance in the human population, our

**Data Availability Statement:** Short-read sequence data for all 154 isolates used in this study are deposited in the NCBI SRA and can be found associated with BioProject PRJNA675772 available on: https://www.ncbi.nlm.nih.gov/bioproject/PRJNA675772. Computer code The code used in this study is available in the following GitHub repository: https://github.com/tan0101/EcoliWGS-PLOSCompBiology-2022.

**Funding:** ZP was supported by the Ministry of Science and Technology of P. R. China under Grant Key Project of International Scientific and Technological Innovation Cooperation Between Governments (number 2018YFE0101500) and TD, MB, YH and DR were supported by InnovateUK grant [104986], FARMWATCH: Fight AbR with Machine learning and a Wide Array of sensing TeCHnologies. The funders had no role in study design, data collection and analysis, decision to publish, or preparation of the manuscript.

**Competing interests:** The authors have declared that no competing interests exist.

findings provide novel insights into the broader epidemiology of antimicrobial resistance in livestock farming. Moreover, our original data analysis method has the potential to uncover AMR transmission pathways when applied to the study of other pathogens active in other anthropogenic environments characterised by complex interconnections between host species.

## Author summary

Livestock have been suggested as an important source of antimicrobial-resistant (AMR) *Escherichia coli*, capable of infecting humans and carrying resistance to drugs used in human medicine. China has a large intensive livestock farming industry, poultry being the second most important source of meat in the country, and is the largest user of antibiotics for food production in the world. Here we studied antimicrobial resistance gene overlap between *E. coli* isolates collected from humans, livestock and their shared environments in a large-scale Chinese poultry farm and associated slaughterhouse. By using a computational approach that integrates machine learning, whole-genome sequencing, gene sharing network and mobile genetic elements analysis we characterized the *E. coli* community structure, antimicrobial resistance phenotypes and the genetic relatedness of non-pathogenic and pathogenic *E. coli* strains. We uncovered the network of genes, associated with AMR, shared across host species (animals and workers) and environments (farm and slaughterhouse). Our approach opens up new avenues for the development of a fast, affordable and effective computational solutions that provide novel insights into the broader epidemiology of antimicrobial resistance in livestock farming.

## Introduction

A recent study showed that some environments can promote the enrichment of antimicrobial resistance genes (ARGs) and exchange between environmental microbiota, human commensals, and pathogens [1]. One such environment is poultry production farms, which have been shown to act as reservoirs of antimicrobial resistance (AMR), with multidrug resistance found in bacteria, especially *Escherichia coli*, from both healthy and diseased poultry [2–6]. This is particularly true for China, where intensive farming practices have the potential to act as source of emerging resistance because of the large quantity of antimicrobials used for both livestock health and growth promotion purposes, creating a selective pressure for resistant bacteria to emerge, enrich and spread [7–9].

Evidence has been given, documenting the spread of ARGs and their bacterial hosts, especially *E. coli*, between chickens and humans in poultry farming [2,10]. The importance of poultry as a source of antimicrobial genes in humans has been demonstrated by studies finding that the Chinese, European and American human microbiome and ARG profiles share more genes with chickens compared to the gut of pigs or cattle [11].

Characterizing the bacterial community structure and AMR gene exchange with *E. coli* in poultry is of interest for several reasons. First, there is suggestive evidence that animals may serve as reservoirs for *E. coli* found in humans [12,13]. Second, in addition to being an important commensal of human and mammalian intestinal microflora, the species *E. coli* contains many pathotypes that are also highly pathogenic and potentially fatal to humans, with some strains having a minimum infective dose as low as just ten cells [14]. Third, *E. coli* are

widespread in different habitats and almost ubiquitous in the avian and mammalian gut microbiota, and, as a result of their transmissible plasmids, able to acquire antimicrobial resistance. Evidence has shown that the spread of resistant bacteria, including *E. coli*, and their AMR repertoire takes place by direct contact and interaction between humans and the microbiota of the surrounding environment [15]. This dissemination is facilitated by horizontal gene transfer (HGT) mediated by mobile genetic elements (MGEs) transferred between multiple pathogen hosts [1,10,16] and phylogenetically closely related bacteria including commensals such as *E. coli* [11].

It is estimated that more than one billion people work in the agriculture sector worldwide, with poultry being the second most widely produced and eaten source of meat in the world. Therefore, AMR originating from environmental reservoirs and farm animals represents a high risk for man as it may be exchanged with human pathogens and commensals. Understanding if antimicrobial-resistant pathogens causing human disease and/or their resistance genes are commonly acquired from livestock, and how and to what extent AMR transfer occurs in these settings will improve our ability to track and assess the risk of emergence of zoonotic infections with the associated risks of reduced tractability by chemotherapy.

Several recent studies have undertaken to characterise the genome of *E. coli* isolates in relation to AMR in poultry farms [17–19]. In addition, several papers have used machine learning to predict AMR phenotypes from both *E. coli* isolates [20–26] and other bacteria [27–30]. To our knowledge this is the first One Health comprehensive analysis and prediction of *E. coli* genome features related to AMR phenotypes, within an intensive poultry farm and connected slaughterhouse.

The aims of this study were to: (i) characterize the *E. coli* community structure, antimicrobial resistance phenotypes and the genetic relatedness of non-pathogenic and pathogenic *E. coli* strains in a large-scale commercial poultry farm in China; (ii) unravel the network of genes, associated with AMR, shared across host species (animals and workers) and environments (farm and slaughterhouse); (iii) identify genes that cross habitat boundaries via their association with mobile genetic elements. More generally, the study aimed to develop an original data analysis pipeline, combining omics, machine learning, gene sharing network and mobile genetic elements analysis, useful to uncover genetic signatures of resistance to antimicrobials and map potential transmission pathways amongst host species and interconnected environments.

## Results

### Poultry farm and slaughterhouse environments contain a diverse array of both non-pathogenic and pathogenic multidrug resistant *E. coli* with phylogenetic intermixing among livestock, humans, meat and environment

We collected and sequenced a total of 154 *E. coli* isolates from two sites, a commercial poultry farm in Shandong province, China, and the contracted slaughterhouse in the same province. Isolates were collected from three hosts: chickens (n = 82; faecal, cloacal, caecal and carcass), humans (n = 58; faecal, hand and nasal) and environment (n = 14; soil, water, feed) (Fig 1A and S1 Table). Samples were taken over a two independent 6-weeks broiler production cycles (first cycle denoted by *t*, second cycle denoted by *l*), with samples taken at three time points: $t_1/l_1$ –mid-cycle (3 weeks old chickens), $t_2/l_2$ –fully grown (6-week-old chickens) and $t_3/l_3$ – end-of-life (6 weeks + 1day, chickens after slaughter). In total 114 isolates were taken from the farm location (67 chicken, 11 environment and 36 human) and 40 isolates were taken from the slaughterhouse (15 chicken, 3 environment and 22 human). To compare the genetic relatedness of the cohort, a maximum likelihood phylogenetic tree for the 154 genomes based on the

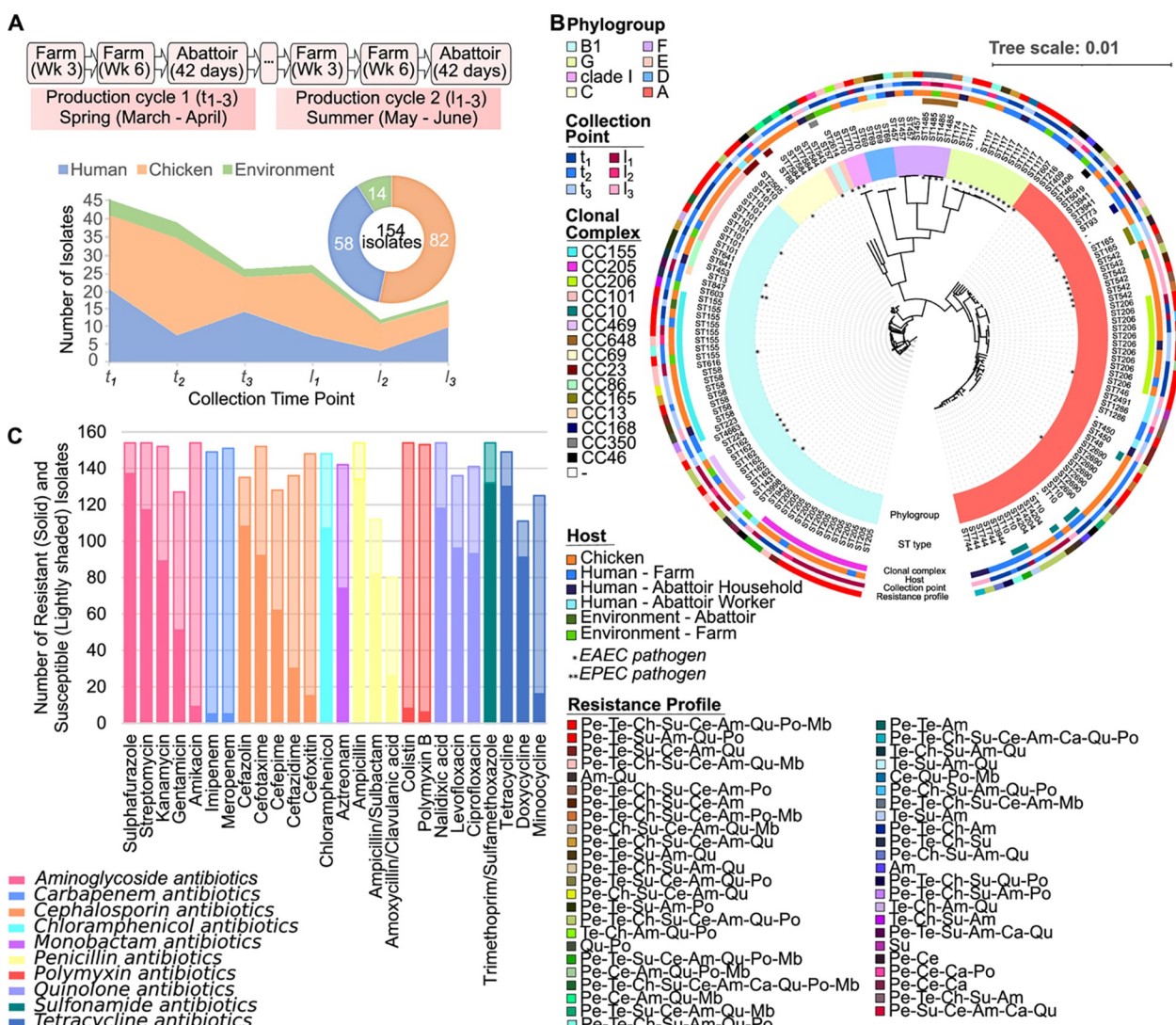

**Fig 1. A wide array of non-pathogenic and pathogenic *E. coli* with phylogenetic intermixing, and high prevalence of shared multidrug resistances amongst livestock, humans, meat and environment are present in a poultry farm and the contracted slaughterhouse.** (A) Overview of the study design and cohort information. 154 human, animal, meat and environmental samples were collected at each of three time points (denoted t1-3 and l1-3) over two independent 6-weeks broiler production cycles over one farm as well as the abattoir linked to this farm in China. The total number and type of isolates cultured and sequenced by host and host body site are shown. (B) Maximum likelihood phylogenetic tree of the whole cohort based on core genome of the 154 isolates with recombination correction, cultured from the human, animal and environmental and meat samples collected from the farm and slaughterhouse. Phylogroups (inner ring), sequence types, clonal complex, collection time points, host type and resistance profile (outer ring) are shown around the outside of tree. EAEC and EPEC isolates indicated by ∗ and ∗∗ respectively. (C) Numbers of susceptible and resistant isolates, based on antimicrobial susceptibility testing by broth microdilution, for each of a panel of 26 antimicrobials encompassing nine different antimicrobial groups.

core genome (2788 genes, present in ≥ 99% of isolates) was constructed. The tree coupled with the *in silico* typing demonstrated high genetic diversity overall with phylogenetic inter-mixing between isolates from human and livestock (Fig 1B). The cohort spanned six of the seven common phylogroups (A, B1, C, D, E and F) as well as the more recently identified G and clade I groups (S1 Table). Phylogroups were not significantly associated with a host (chi-squared test with p-value$_\text{Bonferonni}$ < 0.05) or farm/slaughterhouse environment (chi-squared p-value$_\text{Bonferonni}$ < 0.05). Multi-locus sequence typing revealed diversity with 52 different

sequence types across the cohort (S1 Table). 60% of the isolates could be further grouped into clonal complexes with CC155 (n = 17), CC205 (n = 13) and CC206 (n = 11) the most prevalent. CC155 [31,32] as well as other less prevalent CCs in our cohort have been previously associated with multidrug resistance (CC10 [n = 6] (52), CC469 [n = 6] [33,34], CC648 [n = 4] [3,34] and CC86 [n = 3] [35,36]. Analysis of pathotypes and serotypes (S1 Table) showed that the majority of isolates in our cohort were non-pathogenic bacteria, however, 39 isolates (25.3%) were found to be EAEC strains and 1 was EPEC (Fig 1B). The EAEC isolates were found in all phylogroups but were significantly associated with phylogroup G (chi-squared p-value$_{Bonferonni}$ < 0.0001), sequence types ST117 (chi-squared p-value$_{Bonferonni}$ < 0.0001) and ST542 (chi-squared p-value$_{Bonferonni}$ < 0.015). EAEC isolates were found across all hosts and sources (Fig 1B), however, farmworker nose swabs were significantly associated with these pathogens (chi-squared p-value$_{Bonferonni}$ < 0.0198).

All isolates underwent laboratory testing for resistance to a panel of 26 antimicrobials encompassing 9 classes: beta-lactams, aminoglycosides, chloramphenicol, quinolones, glycopeptides, tetracyclines, sulphonamides, polymyxins and trimethoprim. The proportion of resistance to each antimicrobial ranged from 3% to 89% (Fig 1C, Table 1). Resistance to antibiotic of last resort colistin/polymyxin was found in only 5% of isolates (n = 8) sourced from

**Table 1. Number of resistant and susceptible isolates included in this study.**

| Antibiotic | Abbreviation | Number of Resistant Isolates | Number of Susceptible Isolates |
|---|---|---|---|
| Amikacin | AMI | 9 | 145 |
| Amoxycillin/Clavulanic acid | AMC | 26 | 54 |
| Ampicillin | AMP | 134 | 20 |
| Ampicillin/Sulbactam | AMS | 82 | 30 |
| Aztreonam | AZM | 74 | 68 |
| Cefazolin | CFZ | 108 | 27 |
| Cefepime | FEP | 62 | 66 |
| Cefotaxime | CTX | 92 | 60 |
| Cefoxitin | CFX | 15 | 133 |
| Ceftazidime | CAZ | 30 | 106 |
| Chloramphenicol | CHL | 107 | 41 |
| Ciprofloxacin | CIP | 93 | 48 |
| Colistin | CT | 8 | 146 |
| Doxycycline | DOX | 91 | 20 |
| Gentamicin | GEN | 51 | 76 |
| Imipenem | IMI | 5 | 144 |
| Kanamycin | KAN | 89 | 63 |
| Levofloxacin | LEV | 96 | 40 |
| Meropenem | MEM | 5 | 146 |
| Minoocycline | MIN | 16 | 109 |
| Nalidixic acid | NAL | 118 | 36 |
| Polymyxin B | PB | 6 | 147 |
| Streptomycin | STR | 117 | 37 |
| Sulphafurazole | SUL | 137 | 17 |
| Tetracycline | TET | 130 | 19 |
| Trimethoprim/Sulfamethoxazole | SXT | 132 | 22 |

No statistical differences were found between the number of resistant isolates in the farm and slaughterhouse (chi-squared p-value$_{Bonferonni}$ < 0.05), either overall or by antimicrobial. Of the colistin/polymyxin resistant isolates, all were multidrug resistant to at least 7 antimicrobials with the median number of resistances being 19.5.

chicken, human and environmental hosts. The median number of resistances per isolate was 13 and 96.7% of isolates showed resistance to at least one antimicrobial, with 92.2% having resistance to at least three. Notably, 12.3% of the cohort were resistant to at least 20 antimicrobials, with 23 being the maximum number of resistances for a single isolate indicating extremely high levels of antimicrobial resistance among the *E. coli* present in this farming environment. The most prevalent resistance profile (n = 47) in the cohort was penicillins, aminoglycosides, mono-beta-lactams, cephalosporins, chloramphenicol, quinolones, tetracyclines, sulphonamides and polymyxins. This profile was found present in different ST type clusters in the tree (ST205, ST155, ST101, ST117, ST206, ST2690) with three of these containing chicken and human isolates (ST205, ST155, ST206) as well as one containing human and an environmental isolate (ST117), see Fig 1B.

## Network analysis based on pairwise single nucleotide polymorphisms (SNPs) alignment highlights complex pattern of highly related isolates crossing host boundaries

To assess the relatedness of isolates across our cohort we compared the number of core gene SNPs per isolate in a pairwise manner. Across the cohort the median SNP difference was 18386 (range 0–52377, IQR 22240). Setting a maximum threshold of 15 SNPs as previously done in a similar study [37], we assessed the relatedness of the isolates in our cohort (S1 Fig). The resulting network showed several host-specific clusters of chicken- or human-only samples. The human-only clusters were non-pathogenic whilst two chicken-only clusters contained pathogenic isolates only, and a further two showed pathogenic and non-pathogenic isolates intermixed. Interestingly, the network also showed inter-host clusters. Specifically, three clusters of chicken and human samples (maximum SNP differences of 2, 5 and 8 SNPs; two non-pathogenic and one mixed); two clusters of human and environmental samples (0 core SNP differences in each cluster; one pathogenic and one non-pathogenic); three clusters of chicken and environment (maximum SNP differences of 0, 2 and 5, two pathogenic and one mixed) and finally two clusters with all three hosts intermixed (maximum SNP differences of 1 SNP per cluster; one mixed and one non-pathogenic). This close genetic relatedness (0 to 8 SNPs) among livestock, human and environmental isolates highlights the potential for transmission of pathogenic and non-pathogenic *E. coli* bacteria within this environment.

## Machine learning identifies with high accuracy known and potentially novel genetic determinants of resistance to twenty-one different antimicrobials, discriminating drug-resistant and susceptible strains in the cohort

To test whether the antibiotic resistant/susceptible phenotypes could be explained by known AMR-associated genes (including genes that are known to confer antimicrobial resistance based on their annotations and well-known AMR genes) present in the cohort, the isolates were scanned for sequence matches to genes in AMR gene databases (CARD [38], ResFinder [39], ARG Annot [40], NCBI AMRfinder [41]). The Jaccard/Tanimoto coefficient of similarity between resistance phenotypes for each antibiotic and the known AMR-associated genes was calculated in a pairwise manner. Values spanned the range -0.13 to 0.53, where 1.0 would indicate a perfect positive association and -1 would indicate a perfect negative association. These results indicated that no single known AMR-associated gene could be found that explained the resistance phenotype observed for each antibiotic. The strongest association (0.53) was found between the gene *rmtB* and resistance to amikacin, an association that has been observed before in China [42]. The lack of a simple association between known AMR-associated genes

and resistance together with the genetic relatedness found in our cohort motivated further analysis to identify genetic associations with our observed phenotypes.

To this end we applied a machine learning approach to specifically deduce the relationship between genome content and AMR profiles of each isolate to each one of the antimicrobials. To avoid any bias in the supervised learning approach we first tested for the presence of clonality in the population. To do this we calculated the standard association index ($I^S_A$) [43]. The calculated $I^S_A$ value was 0.2497 (p-value < 0.00001) at whole cohort level and 0.1111 (p-value < 0.00001) as ST type level (one isolate used to represent each of the 52 STs) meaning no linkage disequilibrium was detected indicating the absence of clonality in the *E. coli* population [44].

Ten supervised learning classifiers, logistic regression, linear support vector machine and radial basis function kernel support vector machine (RBF-SVM), extra tree classifier, random forest, adaboost, xgboost, naïve bayes, linear discriminant analysis and quadratic discriminant analysis, were used to predict susceptible and resistant strains for each of the 26 antimicrobials. All genome sequences were processed to give overlapping 13 base pair k-mer features related to each sequence. Five antibiotics (amikacin, polymyxin B, polymyxin E, imipenem and meropenem) did not have enough samples in one class to allow cross validation and SMOTE and so were not taken further. To reduce the number of features analysed by the classifiers a two-step feature selection method was applied: firstly, k-mers with a p-value greater than 0.05 according to the chi-square test were discarded [45,46] and then those remaining were used as input to an Extra Tree Classifier. K-mers with a Gini feature importance above the overall mean were selected [47,48].

To ensure robustness all models were trained and tested in 30 independent runs, with the performance metrics given as the mean of all runs. To verify which classifier performs better out of the 10 classifiers studied we applied a Friedman F-test with Nemenyi post hoc test. With 10 classifiers and 21 antibiotics, the Friedman test is distributed according to the F distribution with 21−1 = 20 and (10−1)×(21−1) = 180 degrees of freedom. The critical value of F(9,180) for α = 0.0001 is 4.0366. For all the metrics studies (AUC, accuracy, sensitivity, specificity, precision and Cohen's Kappa), the $F_F$ statistics null hypothesis was rejected with a confidence level of 99.99%. The null hypothesis states that there are no differences between the average ranks of each classifier over the 21 antibiotics. After, the Nemenyi post-hoc test was performed, and the critical difference diagram was set at 3.407 with a confidence level of 99%. According to S2 Fig, the classifier RBF-SVM had the best rank for the metrics AUC, accuracy, sensitivity, Cohen's kappa, second best for precision and third best for specificity. Nonetheless, according to the Nemenyi test, for precision and specificity, RBF-SVM is statistically equivalent when compared to the best ranked classifier in these two metrics. Therefore, the RBF-SVM classifier was selected for this study. Twenty-one antimicrobial models achieved high performance scores (Fig 2 and S2 Table).

Ampicillin, cefazolin, and tetracycline achieved an AUC score higher than 0.99, with 17 antimicrobials achieving an AUC between 0.9 and 0.99. Only streptomycin achieved an AUC score below 0.9. The antimicrobial with the highest AUC score (0.998 ± 0.002) was cefazolin, it achieved also an accuracy of 0.977 ± 0.009, sensitivity of 0.982 ± 0.01, specificity of 0.958 ± 0.037 and a Cohen's Kappa score of 0.929 ± 0.029. Similarly, the precision was also high with values ranging from 0.706 ± 0.107 to 0.991 ± 0.03 and 11 models having a mean precision over 0.94.

To verify if the number of samples for each antibiotic model was large enough for the test set to be representative, we employed a wrapper backward selection (WBS) in terms of the samples using two different analyses: (i) firstly, we applied the WBS (see S3A Fig), to the original pipeline used in this work (i.e. nested cross validation with an RBF-SVM as the classifier

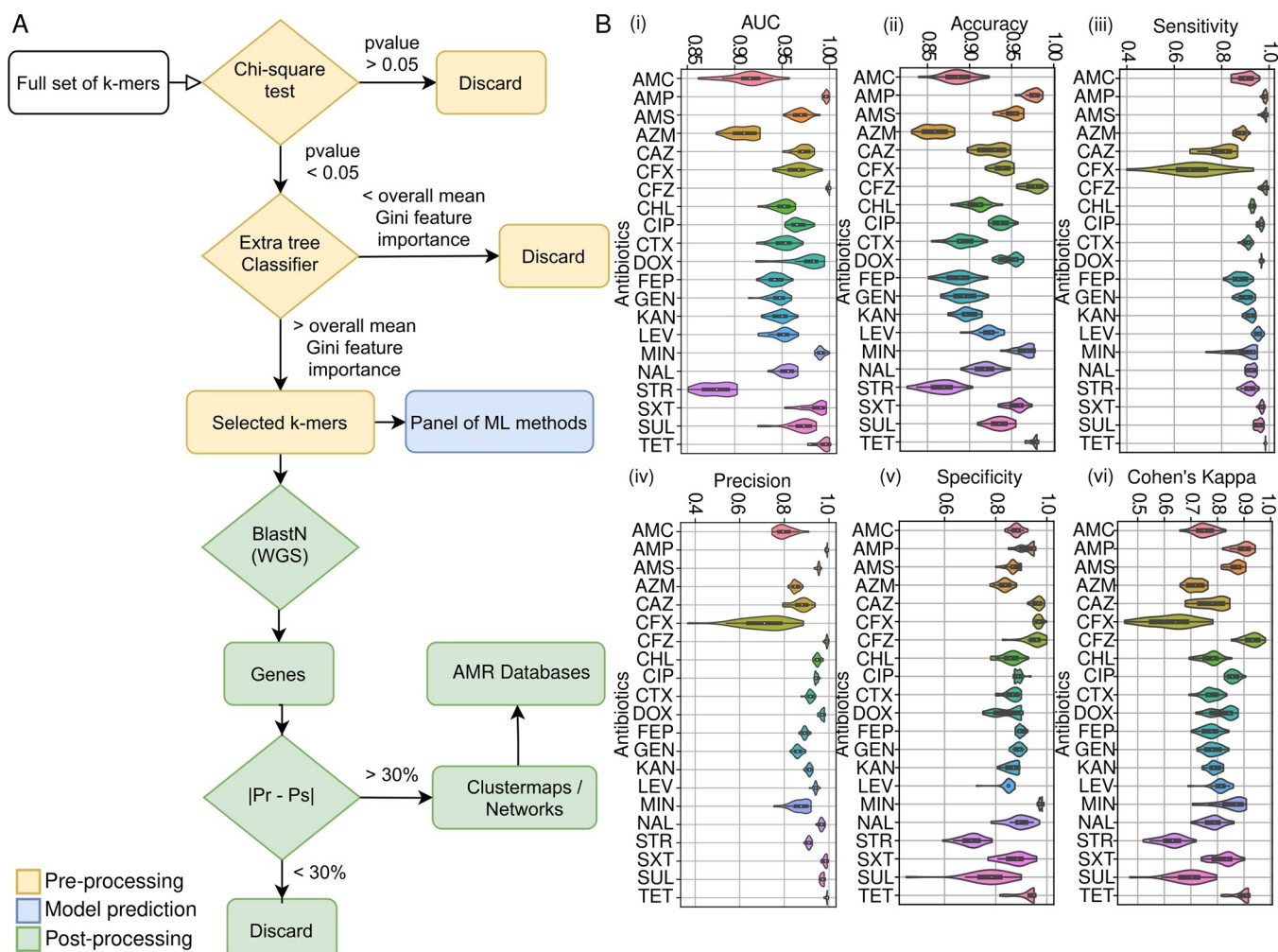

**Fig 2. Supervised machine learning prediction of antimicrobial resistance signature profiles to 21 antimicrobials in the *E. coli* cohort.** (A) Flow diagram showing the machine learning pipeline including feature selection (yellow), classification (blue), post-processing (green). (B) Prediction performance results of the RBF SVM classifier that achieved the best performance amongst the three investigated is shown. Five performance indicators have been used to evaluate the classification: i) Area under the curve AUC, ii) Accuracy, iii) Sensitivity, iv) Precision, v) Specificity, and vi) Cohen's Kappa value from 30 training runs for each antimicrobial. The scores for each performance metric are indicated in the Y axis. Predictive models were generated to classify the resistance vs. susceptibility profiles of twenty-one different antimicrobials (X axis): amoxycillin/clavulanic acid (AMC), ampicillin (AMP), ampicillin/sulbactam (AMS), aztreonam (AZM), ceftazidime (CAZ), cefoxitin (CFX), cefazolin (CFZ), chloramphenicol (CHL), ciprofloxacin (CIP), cefotaxime (CTX), doxycycline (DOX), cefepime (FEP), gentamicin (GEN), kanamycin (KAN), levofloxacin (LEV), minocycline (MIN), nalidixic acid (NAL), streptomycin (STR), trimethoprim/sulfamethoxazole (SXT), sulfisoxazole (SUL) and tetracycline (TET).

and a SMOTE approach during the training phase to balance the classes for the training of the classifier (as used in our ML pipeline), 5 iterations of the nested cross validation were used; (ii) while, for the second analysis, we first performed a SMOTE approach as a pre-processing step to oversample the minority class and then applied the WBS method on the oversampled data, and analysed using 5 iterations of a nested cross validation with a RBF-SVM as the classifier (S3B Fig). S3A Fig shows that for 4 out of 21 antibiotics (AMP, TET, CFX and MIN) the WBS finished after a few iterations due to the small number of samples in the minority class. Nonetheless, according to the learning curves of the other 17 out of 21 antibiotics on S3A Fig the training and test datasets are representative for the initial number of samples in each antibiotic; with a reduced number of samples the majority of the antibiotics have a decrease in the test

fold performance, indicating that a reduced number of samples in this case can cause an over-fitting of the classifier, since the test fold is not representative anymore. The curve of the test performance for all models is reaching an asymptote indicating that increasing sample sizes further would be unlikely to significantly improve model performance. Moreover, on S3B Fig, which creates synthetic data for the minority class during the pre-processing, reveals a similar result to the first approach in S3A Fig. The shape of the learning curves is kept similar between the two studies and it confirms for all the antibiotics that an increase in the number of samples would not have a big impact in the performance when compared to the original number of samples.

Since our goal was not only to predict resistance with high accuracy, but to also extract key insights from the data, we asked whether the uncovered genetic features are either true determinants of AMR or possible artefacts of the statistical learning algorithm. Therefore, we mapped the k-mers back to the sequences of the whole cohort to identify the genes to which the k-mers from each model were related to. Across all models the k-mers mapped back to 2949 unique genes of which 476 were known AMR-associated genes (held in the CARD [38], ResFinder [39], mutationDB [49], NCBI AMRFinder [41] or ARG-Annot [40] databases). To further identify the most significant genes we identified those with the greatest difference between the percentage of resistant isolates identified as having a k-mer hit to the gene (Pr) and the percentage of susceptible isolates identified as having a k-mer hit to the gene (Ps). Genes with a difference |Pr–Ps| in frequency equal or higher than 30% over the whole cohort, were selected and identified as important to differentiate resistant and susceptible isolates (S3 Table). Altogether, 361 genes were significantly differentiated between resistant and susceptible isolates. Of these 361 genes, 35 were present in three or more antibiotic classes in the predictive antibiotic models (S3 Table). These 361 genes were analysed for their distribution and frequency among the different hosts in both the farm and slaughterhouse environments (S3 Table).

We discuss the results obtained for three antibiotics representing two important classes—beta-lactams (penicillins and cephalosporins) and aminoglycosides—selected because of their clinical relevance in China. Results for the other 18 antibiotics are given in S4 Fig.

The ampicillin/sulbactam (penicillins) model identified 20 genes as significantly separating resistant and susceptible isolates, and of these 18 were mostly present in the resistant isolates (S3 Table). These included six known AMR-associated genes (*lysU* [49], *lptD* [49], *aadA2* [38], *aph(4)-Ia* [38], *floR* [38], *bla*$_{OXA-10}$ [38]). The *floR* gene was present in > 92% of resistant farm samples but in less than 32% of resistant slaughterhouse samples (chi-square *p* value < 0.0001). Of the known AMR-associated genes *aadA2*, *floR* and *bla*$_{OXA-10}$ had multiple k-mers associated with each gene (16, 7 and 4 respectively), statistically unlikely to occur in each gene by chance (binomial test *p* values < 0.0001) whilst the others has only 1 association (mean *p* value = 0.02). Among the 14 genes associated with either a resistant or susceptible phenotype but that are not known to confer antimicrobial resistance based on their annotations (*alx, hcaB, hdfR, hha, nadB, tnpR, traC$_1$, traC$_2$, traV, yciC, iroB, iroN, iucC* and *fes*): four had functions associated with HGT (*tnpR, traC$_1$, traC$_2$* and *traV* [50,51]); and 4 were virulence genes involved in siderophore-mediated iron transport (*iroB, iroN, iucC* and *fes* [52]).

For cefepime (cephalosporins) resistance/susceptible model (S3 Table) 18 genes (*aadA2, floR, bla$_{CTX-M-55}$, cai, cusF, ecpD, exuT$_1$, exuT$_2$, finO, aph(4)-Ia, mdtB, nagE, aph(3")-Ib, papC, tdh, traI, traQ, traS* and *yfeZ*) were significantly associated with the resistance phenotype and a further three with the susceptible phenotype (*mutL, yehL* and *yehA*). Seven genes (*mutL* [49], *aadA2* [38], *aph(3")-Ib* [38], *aph(4)-Ia* [38], *bla$_{CTX-M-55}$* [38], *floR* [38] and *mdtB* [38]) are known AMR-associated genes. Interestingly, *floR* was significantly more present in the resistant chicken isolates, 77%, compared to the resistant human isolates (from farm and

slaughterhouse), 28%, (chi-square p value < 0.001). For the ESBL resistance gene $bla_{CTX-M-55}$ however no statistically significant difference was observed and no co-occurrence of these two resistance genes in the same contig was observed (Jaccard coefficient 0.0). The gene *mutL*, a DNA repair gene, which was the only known resistance gene to be more prevalent in resistant human isolates (67%) compared to chicken isolates (33%), chi-square *p* value = 0.13. The known resistance gene $bla_{CTX-M-55}$ had 17 unique k-mers differentiating the susceptible and resistance phenotypes predicted to be associated with it, unlikely to arise by chance (binomial *p* value < 0.0001), whilst two genes (*aph(3")-Ib and floR)* had between 3 and 6 k-mers (binomial *p* values > 0.0001) and four genes (*mutL, ecpD, aadA2, aph(4)-Ia*) had only a single k-mer related to them. A pattern of higher prevalence in chicken isolates was observed in several of the genes found by the machine learning but not annotated to have AMR function: *nagE, cai, exuT₁, tdh* and *papC* (chi-square p values <0.01). One gene was a known virulence-related pilus gene (*ecpD* [53]). Not surprisingly, none of these genes were found to significantly co-occur with each other or the known AMR-associated genes.

The kanamycin (aminoglycosides) resistance/susceptible model (S3 Table) showed that two genes *tetA* and *rihA* were significantly associated with susceptible isolates whilst 26 were associated with resistant isolates (*aph(3')-Ia, aph(3")-Ib, aph(4)-Ia, $bla_{CTX-M-55}$floRr, $bla_{TEM-1}$, $bla_{OXA-10}$, der, hdfR, mntB₁, mntB₂, mntB₃, dcm, tfaE, traC₁, traC₂, sopB, traD, ylpA, hcaB, traQ, cai, cmi, higB-1, dsbC,* and *repB*). Eight of these were known AMR-associated genes, present in the CARD database (*aph(3')-Ia, aph(3")-Ib₂, aph(4)-Ia, $bla_{CTX-M-55}$, floR, $bla_{TEM}$, $bla_{OXA-10}$* and *tetA* [38]). Of these *aph(4)-Ia* frequently co-occurred with the *aph(3')-Ib* and *aph(3")-Ib* (Jaccard coefficients of 0.51 and 0.20) but no other significant co-occurrence was observed. The genes were generally present in similar proportions of chicken and human isolates. The known AMR genes had significantly more unique k-mers associated with them compared to the genes not known to confer AMR with a mean of 17 compared to a mean of 4 (t-test *p* value < 0.01). All the known AMR genes has a statistically significant number of k-mers, unlikely to occur by chance (binomial *p* value <0.0001 for all except *aph(4)-Ia* with *p* value < 0.01). In addition, several of these genes, were also unlikely to be spurious hits: *der* (*p* value < 0.0001), *hdfR* (*p* value < 0.0001), *mntB₁* (*p* value < 0.0001), *mntB₂* (*p* value < 0.0001), *mntB₃* (*p* value < 0.0001), *dcm* (*p* value < 0.0001), *tfaE* (*p* value < 0.0001), *tetA* (*p* value < 0.0001), *traC₁* (*p* value < 0.0001), *traC₂* (*p* value < 0.0001), *sopB* (*p* value < 0.0001), *traD* (*p* value < 0.001), *ylpA* (*p* value < 0.0001), *hcaB* (*p* value < 0.0001)).

In addition to the k-mer mapping to the annotated genes of the cohort, many hits were found to map to intergenic regions of the DNA sequences. We analysed these hits to identify any patterns and found that the k-mers mapped to 14 different transcription factor binding sites (*arcA, argR2, crp, fis, fur, ilvY, lexA, lrp, nagC, phoB, purR, rpoH2, rpoN* and *tus)* where there was greater than 30% difference between hits to resistant and susceptible isolates, S5 Fig. Of particular interest, the transcription factors *rpoH2* and *crp* were significant in 5 different antibiotics: ampicillin, ampicillin/clavulanic acid, ampicillin/sulbactam, levofloxacin and sulfisoxazole; ciprofloxacin, chloramphenicol, ampicillin/sulbactam, doxycycline and cefoxitin, respectively. Both these transcription factors are global transcriptional regulators binding regulating many genes. The well-studied regulator Crp (cyclic-AMP receptor protein) regulates over 180 different genes, primarily in energy-metabolism pathways [53]. Whilst the alternative sigma factor RpoH is involved in regulating the cellular response to heat shock, increasing bacterial survival at high temperatures [54]. The gene *purR*, found in 4 antibiotic resistance/susceptible predictive models (cefepime, aztreonam, cefotaxime and cefoxitin) is involved in the regulation of genes in the purine metabolic pathway. Importantly, purine metabolism and synthesis pathways have been previously found to be enriched in multidrug resistant bacteria [22]. The transcription factor *fur* present in the cefotaxime, ampicillin, ampicillin/clavulanic acid

and doxycycline models, represses genes involved in high affinity iron sequestration important in virulence [55] and is important in conferring protection from oxidative stress [56]. The gene *tus*, involved in the termination of DNA replication, was significant in doxycycline, ampicillin/sulbactam, minocycline and streptomycin. All other transcription factors were significant in 3 or fewer models.

## Network analysis powered by the machine learning uncovers communities of similar isolates and reveals ARGs sharing between hosts

To map the reservoirs of possible antibiotic resistance related genes that pose the highest risks of horizontal gene transmissibility across humans, animals and their environment, we built gene sharing networks and a hierarchically clustered heatmap (clustermap) for each one of the 21 selected antimicrobials (Figs 3, 4 and S4). The networks were analysed to assess modularity and betweenness centrality. In gene sharing networks, nodes with higher betweenness centrality are potentially more interesting as they are involved in a larger number of associations.

In our dataset, we found an extensive network of genes correlated with AMR phenotypes shared between *E. coli* communities of humans, animals, meat and the environment, that was not seen in the SNP network analysis.

The ampicillin/sulbactam gene sharing network (Fig 3A) showed four different communities. Communities C0, C2 and C3 contained most resistant isolates (86%) whilst community C1 contained most of the susceptible isolates (83%), almost all of which were from human hosts (18/20). The three resistant communities found were clearly separated from the susceptible community. Samples from resistant community C0 were primarily chicken and environmental samples. Six resistant human isolates (nose and hand samples) were present in C2 and of these, all except one were identified as pathogenic EPEC or EAEC strains. These were intermixed with chicken isolates that were either pathogenic or non-pathogenic. Only 40% (16/40) of human isolates were resistant to ampicillin/sulbactam compared to 95.1% (58/61) chicken isolates, and of the human isolates hand and nose swabs were more often resistant than faecal isolates (56% compared to 27%). The presence of the genes *lysU* and *lptD* and the absence of the genes *floR* and *hdtR* were important in differentiating susceptible from resistant isolates (Fig 4A). The *floR* gene is known to be involved in sulphonamide (sulfathiazole) and bicyclomycin resistance [57,58], and was present in all resistant communities. The analysis of the communities C0 and C2 indicates the co-presence of *aadA2* (streptomycin resistance), $bla_{OXA-10}$ (beta-lactamase) and *aph(4)-Ia* (phosphorylates the antibiotic hygromycin B) which are all known AMR-associated genes, however these are not co-occurring in the same sequence contigs. In communities C0 and C3, among the genes not annotated as conferring AMR we observed the co-presence of the following virulence genes: *iroB, iucC, iroN* and *iroD*, of which all except *iucC* frequently co-occur on the same contig. The five nodes with highest betweenness centrality were found to be three chicken isolate (one caecal, one faeces and one anal swab) and two human isolates (hand swabs from the farm and slaughterhouse).

The cefepime network (Fig 3B), contains three communities (C2, C3 and C4) where most of the isolates (humans, chicken, and environment) show resistance to this antibiotic (77%), and two communities (C0 and C1) where most isolates are susceptible (76%). The two primarily susceptible communities are clustered together, but the resistant communities show greater separation. Community C4 was populated with mostly pathogenic samples (13/18), indicating the higher relatedness of these isolates. According to the clustermap analysis (Fig 4B), the AMR-associated genes (*aph(3")-Ib, floR, mdtB* and $bla_{CTX-M-55}$) are clustered together; with *aph(3")-Ib* and *floR* presenting mainly in C2 and C3 and $bla_{CTX-M-55}$ and *mdtB* present across communities C2, C3 and C4. Additionally, the resistant and susceptible isolates also differ by

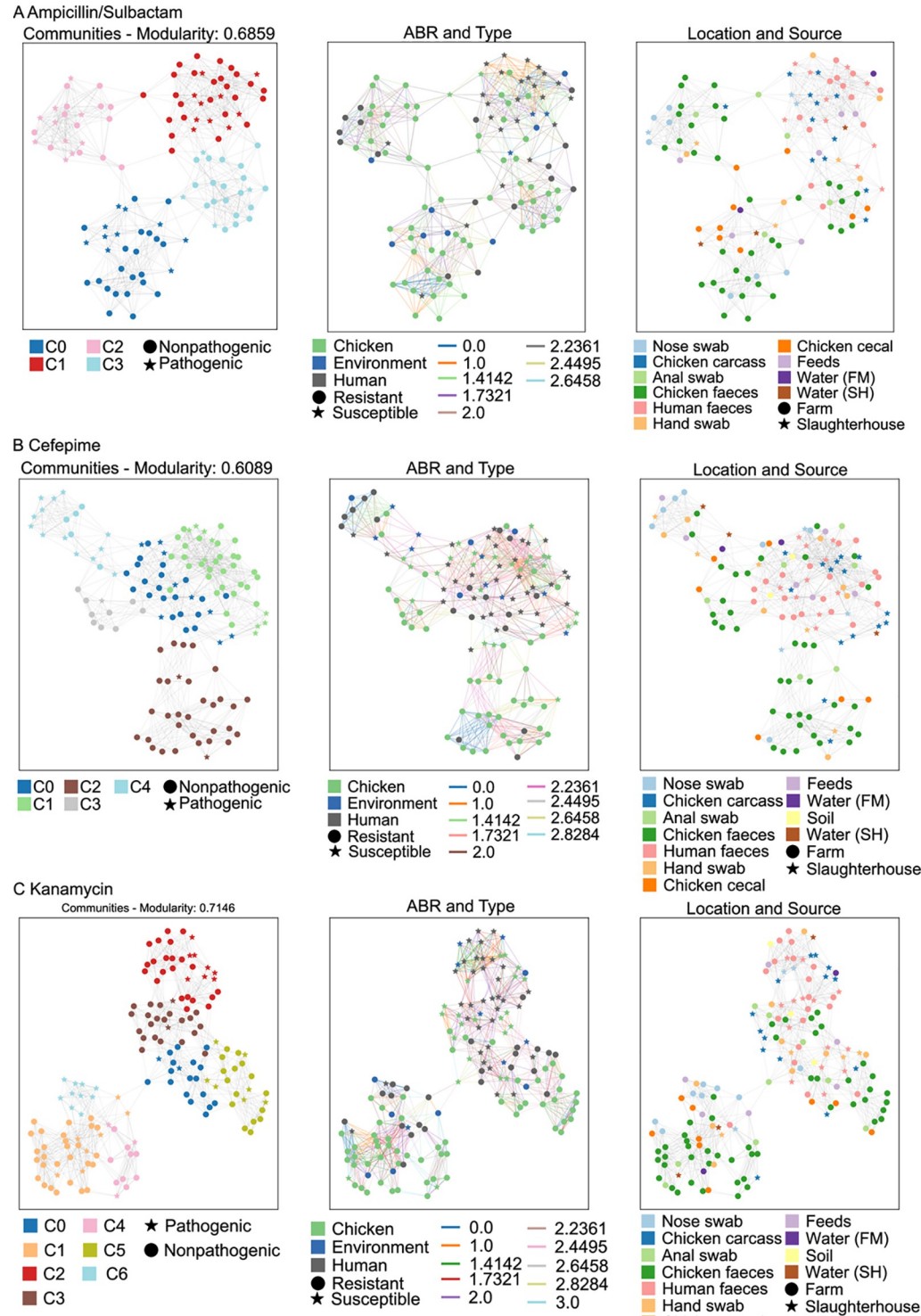

**Fig 3. Networks of AMR-associated genes shared across host species (animals and workers) and environments (farm and slaughterhouse) for three antibiotics.** The AMR-associated gene sharing networks for: (A) ampicillin/ sulbactam, (B) cefepime, and (C) kanamycin are shown. For each network: (i) Panel on the left, indicates the communities and their respective numbers found using Louvain heuristics. Each community is indicated with a distinct colour. Each community is a set of nodes or clusters, which are densely and connected with statistical significance and identically coloured. For each network the node represents a sample and is shown in a distinct colour and shape depending on the metadata of the sample (e.g., source, AMR profile, location). In all panels the nodes are separated

according to the Euclidean distance between the isolates; the central panel shows the AMR phenotype (resistant or susceptible) as the shape of the node and the source of the samples (human, chicken or environmental) as the colour of the node. An edge represents the Euclidean distance between two samples, and it is coloured according to the associated statistical value. The panel on the right indicates the location of the sample (farm or slaughterhouse) as the shape of the node and the type of sample as the colour of the node.

the presence of the genes *yehL* (ATPase) and *yehA* (fimbrial-like adhesin) genes in susceptible isolates. For cefepime, the five nodes with highest betweenness centrality were four chicken samples (one carcass, two faeces and one anal swab) and one environmental (slaughterhouse water) which indicates that the main hotspot is the chicken samples.

In the kanamycin network (Fig 3C) there are three close communities mostly containing susceptible isolates (C0, C2 and C3, 78% susceptible) and the other four (C1, C4, C5 and C6) contained predominantly resistant isolates (91% resistant). Community C6 is formed by only pathogens. Interestingly, there are two major clusters for this antibiotic, one mainly related to resistant samples (communities C1, C4 and C6) and the other mainly related to susceptible samples but also containing resistant community C5. The genes $bla_{CTX-M-55}$, *traC*, *repB*, *higB-1*, *dcm*, *aph(4)-Ia*, $bla_{TEM-1}$, *dsbC*, *hcaB* and $bla_{OXA-10}$, are absent in community C5 but present in the other resistant communities (Fig 4C), with four of them known to confer antimicrobial resistance ($bla_{CTX-M-55}$, $bla_{OXA-10}$, $bla_{TEM-1}$ and *aph(4)-Ia*). The other known AMR-associated genes, *aph(3')-Ia*, aph(3")-*Ib* and *floR* were found present in all the resistant communities. We hypothesise that sharing of AMR-associated genes uncovered by ML is more frequent between the chicken isolates on the farm, since most human and environmental samples are susceptible. Further, 24 out of 38 slaughterhouse isolates were susceptible to kanamycin, indicating that this antibiotic was not significant in the slaughterhouse. For kanamycin, the five nodes with highest betweenness centrality are from multiple sources (one environmental, two humans and two chicken).

Out of the 361 genes that were selected with a |Pr-Ps| > 30%, 303 genes (278 unique excluding allelic variants) were not annotated as antibiotic resistance genes in public AMR databases. Therefore, to characterize these genes without annotated AMR function GO [59] and KEGG pathways [60] enrichment analyses were conducted (S4 Table and S6 Fig). The GO terms showed enrichment in one molecular function encompassing catalytic activity, and thirteen biological processes: metabolic, cellular, cellular metabolic, organic substance metabolic, monocarboxylic acid catabolic, organic cyclic compound metabolic, cellular aromatic compound metabolic, antibiotic catabolic, 3-phenylpropionate metabolic, 3-phenylpropionate catabolic, primary metabolic, small molecule metabolic, oxoacid metabolic and cellular response to chemical stimulus. In addition, four KEGG pathways were enriched: metabolic, microbial metabolism in diverse environments, phenylalanine metabolism and degradation of aromatic compounds (S4 Table and S6 Fig).

### Identification of the mobile genetic elements associated to the interconnected resistomes and their *E. coli* hosts that are naturally transferred among animal, human and the environment

Although the gene sharing network analysis powered by machine learning found extensive interconnected resistomes among *E. coli* communities of humans, animals and environments within and between the farm and slaughterhouse, it remains unclear whether the uncovered interconnections are mediated by mobile genetic elements. We analysed the MGE content of the isolates in our cohort using the mob-suite package [61], then we grouped the isolates by using hierarchical clustering by replicon type. In total, 153 of the 154 isolates in our cohort

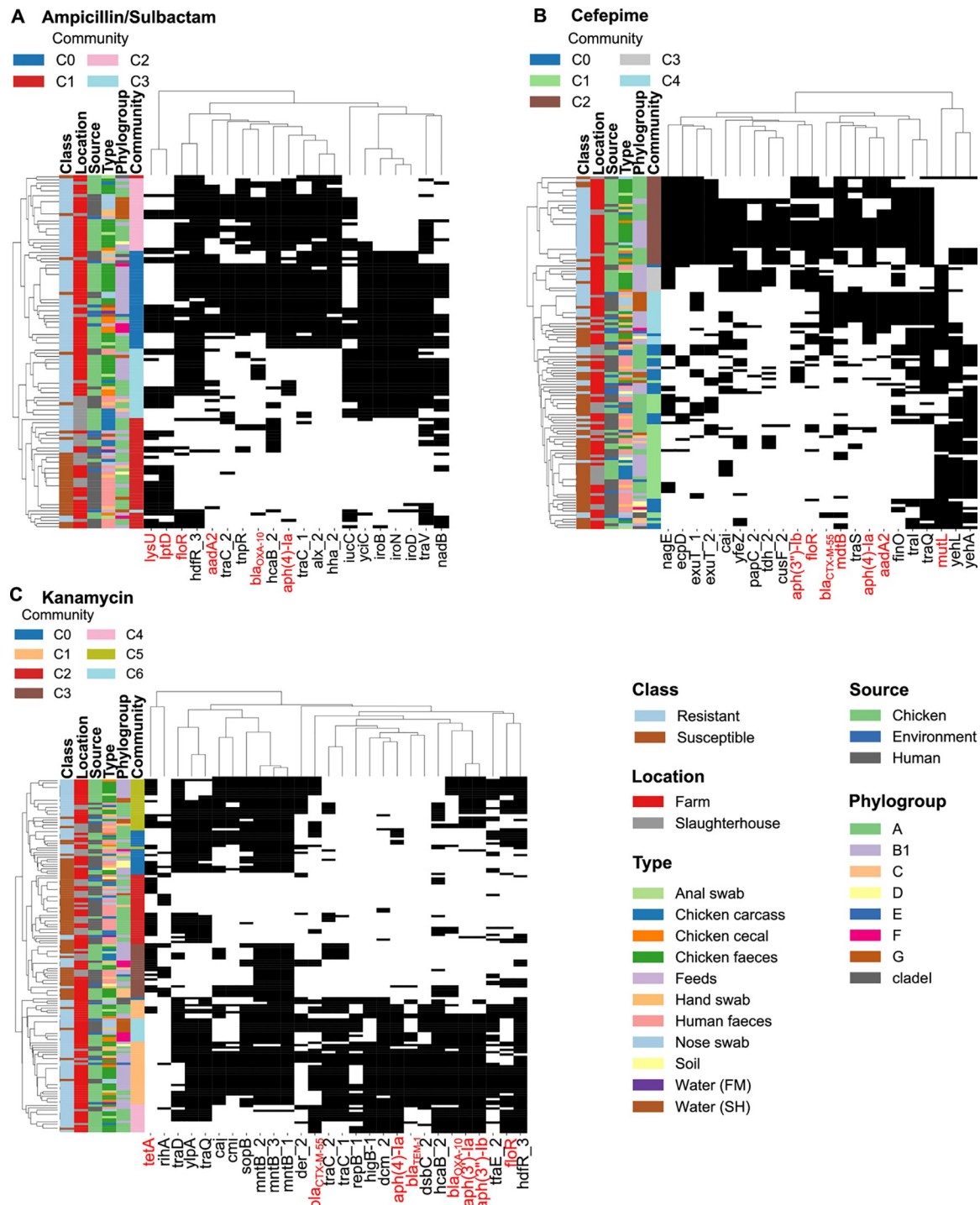

**Fig 4. Hierarchically clustered heatmaps of the AMR-associated genes across the cohort.** Clustered heatmaps showing the genes associated with the significant k-mers from the machine learning and used to build the networks. The antibiotic models shown are: (A) ampicillin/sulbactam, (B) cefepime and (C) kanamycin. The columns on the left show the metadata (Class, Location, Source, Type, Phylogroup, Community number). The presence of the genes (in terms of its related k-mers) is indicated in black, while the absence is indicated in white. Genes known to confer antimicrobial resistance based on their annotations from public databases are highlighted in red.

carried at least one plasmid with 147 of those carrying a mobilizable or conjugative plasmid. Across the cohort, 17.18% of genes in the pan-genome were located on plasmids, with a range of 0–14.6% per isolate. Comparing different hosts, the proportion of genes carried on plasmids in chickens was significantly higher than in human farmworkers (adjusted p-value$_{Bonferroni}$ = 0.00993) and abattoir household members (adjusted p-value $_{Bonferoni}$ = 0.0403) (S7 Fig). The types of plasmid within the cohort were diverse (Fig 5), and did not show any significant clustering by host, collection time point or source.

Next, we tied the above results with the information obtained with machine learning. Replicon types: IncFIA, IncFIB, IncI1/B/O, IncHI1B, IncK, IncR and IncY, previously associated with multidrug resistance [62] carried between 2.57 and 8.45% of the total 4626 genes associated with resistant phenotypes identified by the supervised learning analysis (Fig 6A). Replicon types IncX1 (1.25%) and IncN (1.64%) carried fewer genes, whilst IncI2, IncX4, Col156, ColpVC and Col(MG828), also reported in literature to carry resistance genes [62], carried very few (>1%) of the predicted genes from this study (S5 Table).

In addition to considering the AMR-associated genes uncovered by machine learning, we also annotated the plasmid sequences to assess the gene content of each plasmid replicon. Many known ARGs were carried on multiple plasmids (e.g. *aph(3')-Ia* [n = 6], *aph(3")-Ib* [n = 6],*bla*$_{CTX-M-55}$ [n = 5], *aadA2* [n = 5], *tet*R [n = 5]), S6 Table. Notably, IncI2 also carried resistance gene *bla*$_{CMX-M-97}$ and colistin resistance gene *mcr*-1, an association that have previously been noted to give a fitness advantage via carriage of this gene [63]. Hierarchical clustering of the presence/absence of all genes on each replicon type was conducted to assess the similarity of plasmids from different isolates. We hypothesise that isolates with plasmids clustering with only small differences are indicative of relatively recent transmission of those plasmids between isolates through horizontal transfer. Where those plasmids are present in different host/isolates this would be suggestive of a transmission route for these bacteria.

Although the majority of the plasmids shared predominantly different mobile elements between livestock and humans, we found 17/58 (29.3%) human isolates sharing closely related AMR-associated mobile elements with those found in livestock. Most of the clusters contained different STs which is also indicative of horizontal gene transfer of mobile genetic elements between lineages [37]. Within the IncFIA group there were several small clusters of isolates, Fig 6B. One cluster contained 2 human farmworker nose swabs and an environmental animal feed isolate. Average nucleotide values (ANI) of the WGS of these isolates showed high similarity over >99.995%, compared to a mean ANI value across the cohort of 98.08%. IncFIA is an important replicon for AMR in this cohort and contained several known resistance genes namely: *aad*A2, *tet*A, *tet*R, *cml*A5, *bla*$_{TEM}$, *bla*$_{CTX-M-97}$ and *flo*R. Within the IncI1/B/O replicon group, a chicken carcass sample (non-pathogenic, phylogroup A, ST206) showed high similarity to the hand swab of a household member of an abattoir worker (non-pathogenic, phylogroup G, ST174) (ANI value of WGS = 99.995%) and two chicken faeces samples (non-pathogenic, phylogroups B1 and F, ST101 and ST457 respectively) showed high similarity to a farmworker hand swab isolate (non-pathogenic, phylogroup F, ST457) (ANI values of WGS: 99.995% and 97.1423%). In IncHI1B chicken faeces isolates (non-pathogenic, phylogroup B1, ST205) have clustered with a farmworker nose swab isolate (non-pathogenic, phylogroup B1, ST205) (ANI values of WGS > 99.994%). As with IncFIA, IncHI1B carries many known AMR genes: *cat*, *bla*$_{OXA-1}$, *bla*$_{CTX-M-97,}$ *aad*A2, *cml*A5, *aph(3")-Ib*, *tet*A, and *tet*R. In IncFIB, carrying known resistance genes *tet*A, *tet*R, *cml*A5, *aph(3")-Ib*, *bla*$_{TEM}$, and *flo*R, a chicken carcass plasmid (non-pathogenic, phylogroup A, ST206) clustered with a hand swab of a household member of an abattoir worker (non-pathogenic, phylogroup A, ST206) (ANI value of WGS = 99.996%). All the IncR plasmids (6 isolates, all EAEC positive, phylogroup A, ST542) show high similarity and except for one isolate are all sourced from chicken carcass samples.

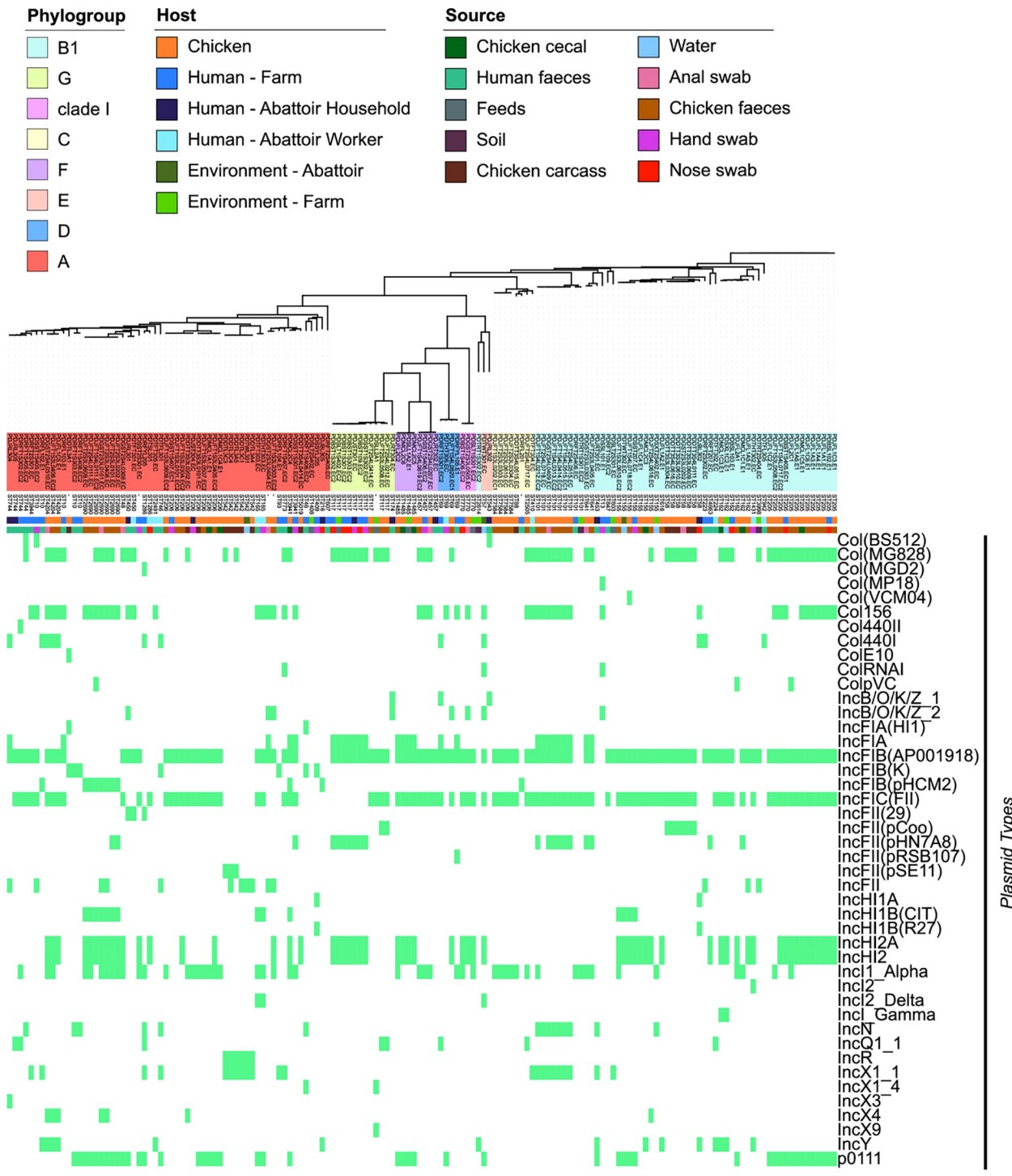

**Fig 5. Presence/absence of different plasmid types in the cohort.** Heatmap showing the presence (green) and absence (grey) of each predicted plasmid replicon type per isolate. Phylogenetic tree (replicated from Fig 1B) is at the top of the heatmap with the phylogroup. Sequence types, hosts and isolate sources are shown below the tree.

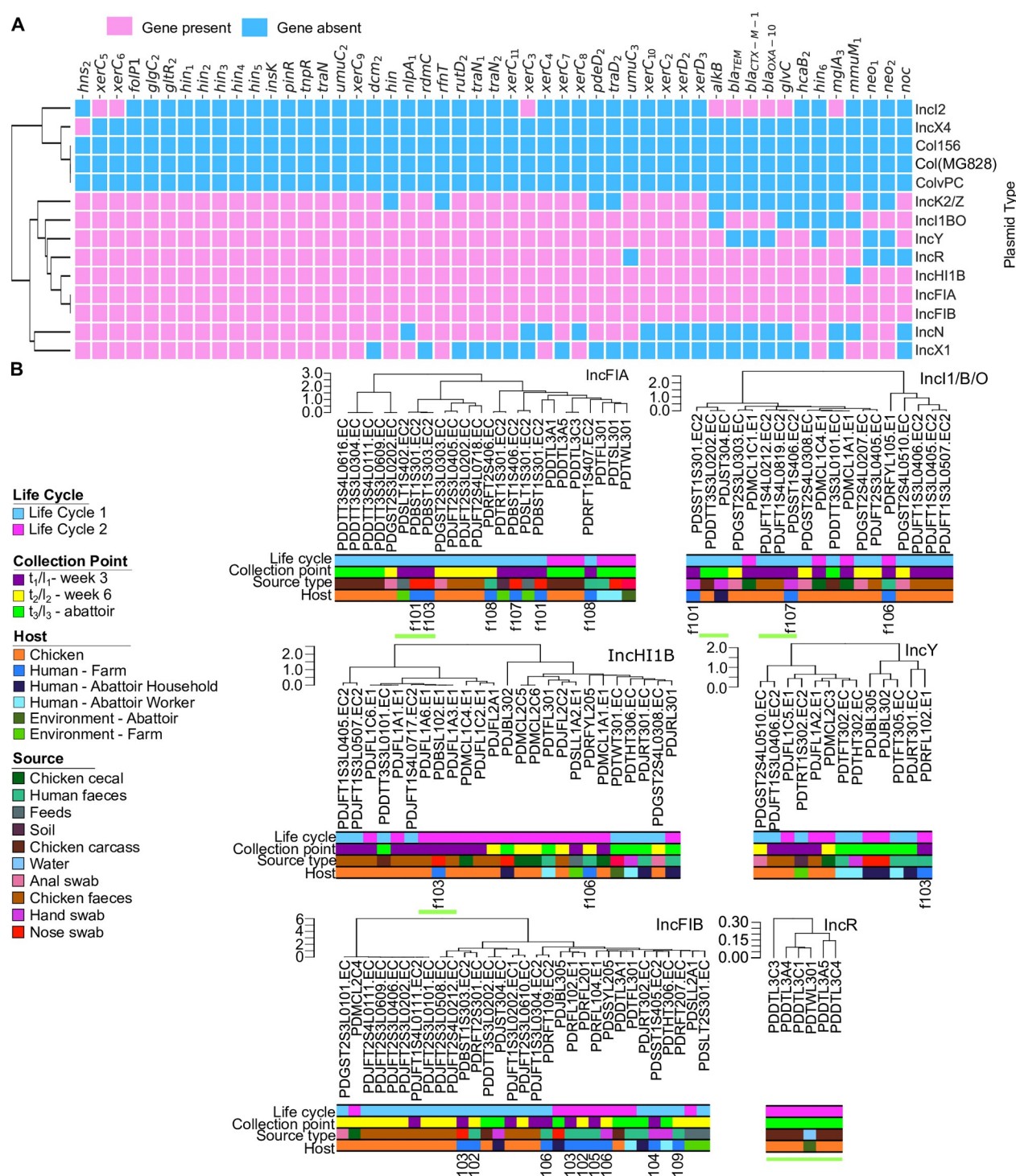

**Fig 6.  Presence of AMR-associated genes predicted by machine learning in plasmids and hierarchical clustering of isolates based on similarity of plasmid type** (A) Heatmap of presence (pink) or absence (blue) of genetic determinants identified by machine learning located on different replicon types of plasmid within the cohort. (B) Hierarchical clustering of isolate plasmid sequences from replicon types with high number of AMR determinants. Clustering was based on gene presence or absence. Green underscores indicate areas of high similarity between isolates from different hosts, suggestive of transmission between hosts.

The other isolate was taken from the water at the abattoir, suggesting a possible transmission route between the chicken carcass and abattoir water system. Other isolates with less significant AMR genes but also sharing plasmids with high pairwise ANI values > 99.996% and showing inter-source clustering, again indicate direct transmission of the bacteria (S7 Fig). We found no evidence of human-to-human bacterial transmission in the farm environment.

## Discussion

Resistance genes are more likely than other genes to cross boundaries of ecological niches and habitats [64]. Characterizing the sources, reservoirs and networks of potential transmission of antimicrobial resistance between human, animal, and environment, is key to isolate the major elements of risk undermining human health, and essential to design efficient and effective interventions opposing the selective forces promoting the enrichment and spread of antimicrobial resistant pathogens.

In this study, we focused on *E. coli* in intensive poultry farming. We developed a computational platform that combines pangenome, machine learning, gene sharing network analysis and the analysis of mobile genetic elements, and applied it to the data collected from a tightly confined and geographical contextualised longitudinal ecosystem consisting of a poultry farm and associated slaughterhouse.

The pangenome properties resulting from our analysis reflect the current understanding of the extent of genetic variability in *E. coli*, with 8 phylogroups and 52 ST types in line with what seen in other studies in China [65–67]. However, 25% of the isolates in our cohort were found to be EAEC. Interestingly, we found these EAEC isolates to be significantly associated with phylogroup G and sequence types ST117 and ST542. Notably, ST117 has been established to have a strong association with avian pathogenic (APEC) *E. coli* [68–70]. In European studies, ST117 has been shown to be a poultry-associated strain, capable of causing human disease [71], and has been associated with multidrug resistance both in Europe and China [72]. Thus, the association of ST117 with human pathogenic strains in this study could be indicative of zoonotic transmission of pathogenic *E. coli*. In particular, the significant association of EAEC isolates with human nose swabs (6 of 11 nose swab isolates across 5 individuals), suggests a direct infection/colonisation route for pathogenic strains within the commercial poultry environment. Our isolates showed similar phylogroups and AMR profiles to a recent very similar cohort (human, broiler and environment isolates taken from poultry farm and market) sampled in Nigeria [17]. Whilst some differences were observed in ST types, with the predominant ST type from our study, ST205, absent in the Nigerian study, there was a large overlap in ST types with 10 of the 14 ST types identified in the Nigerian study also present in ours. As with our study a wide range or resistance genes were observed including colistin resistance genes, *mcr-1*, in our study however, a smaller range of fluroquinolone resistance genes were found (2 compared to 10 in Nigeria).

The association of human with chicken and environmental pathogenic and non-pathogenic *E. coli* strains was further evidenced by the SNP-based network analysis of highly correlated strains, showing both inter- and intra-host clusters and suggesting a possible transmission of *E. coli* between livestock, environment and humans. These results partially contrast with a previous study conducted in the East of England, where limited sharing of *E. coli* strains and AMR genes was found between livestock from multiple farms and human blood stream infections [37]. However, the differences in findings between this and the previously cited study [37] may be justified by our analysis encompassing a different geographical area (with different usage patterns in relation to antimicrobials) [73], and adopting a longitudinal investigation involving two closely interconnected environments (farm and slaughterhouse) and their entire ecosystems of workers and animals.

Of the 361 genes which we found as discriminating the AMR phenotypes with high accuracy, 58 were known AMR-associated genes based on their annotations in public databases. The identification of genes known to confer resistance to the selected resistance phenotypes indicates the robustness of methods employed, as also stressed by Jaillard [74]. The identification of genes that are not known to confer antimicrobial resistance based on their annotations together with known AMR-associated genes demonstrate the potential of machine learning for predicting AMR phenotypes and its ability to generate hypotheses that may increase our understanding of the genetic basis underlying AMR or correlated to AMR in *E. coli* [75–77]. A recent study by Ren [25] also looked at prediction of *E. coli* resistance from human and animal isolates from two datasets, one they term the Giessen data, 987 whole genome *E. coli* sequences they collected as part of their study and a public dataset of 1509 *E. coli* strains taken from Moradigaravand et al. [26]. Ren tested both of these datasets against four antibiotics (ciprofloxacin, cefotaxime, ceftazidime and gentamicin) all of which we also tested allowing comparison between our study and theirs. There were several key differences between our study and theirs: the Giessen data in Ren are described as human and animal clinical samples with no source country given and the public data were retrieved from both human clinical and environmental collections in the UK, Pakistan, Syria, Sweden, the USA, and Belgium. In contrast our data were from interconnected human, animal and environmental sources on a single Chinese poultry farm and connected slaughterhouse. Ren used SNPs (called against *E. coli* K-12 reference) as input to their classifiers, with no feature selection, whereas we used k-mers and performed a two-step feature selection to reduce sample-feature ratio. Ren then used Logistic regression, linear SVM, Random Forest and convolutional neural network classifiers to predict AMR phenotypes. In our study we used 10 classifiers including 3 that Ren used (LR, SVM, RF). Comparing the four antibiotics models across the two studies, our machine learning models achieved equal or higher AUC and precision values compared to all four antibiotic models. The sensitivity of our models was the same or better compared to both the Giessen and public datasets for all antibiotics except ceftazidime, where our sensitivity was 0.81 compared to 0.74–0.90 in the Ren study. For each of their datasets, Ren took the 10 highest-ranking features from each antibiotic and associated them with the genes in the genome to identify the cognate markers of AMR. Excluding known AMR genes, they found 19 genes of interest. Interestingly, 14 out of these 19 genes (*nhaA*, *rlmC*, *fliI*, *pepB*, *prlC*, *sodA*, *murB*, *rluF*, *yjfF*, *treR*, *argI*, *valS*, *fhuF* and *nadR*) were also selected by our machine learning algorithms, and 4 (*sodA*, *rluF*, *treR* and *argI*) were found to be in the 361 most discriminating genes. This overlapping of genes despite the different sample sources and countries (predominantly clinical isolates from humans and animals of international origin, compared to isolates from healthy animals and humans from China in our study) gives further evidence of the robustness of the methods we have used in this study.

Despite the limited number of samples available in our study, when we compared our machine learning performance metrics to published studies with similar and larger *E. coli* datasets we found comparable AUC, sensitivity, and specificity values[20,21,25,26], indicating that the methods employed in this study have not been excessively affected by this limitation. Specifically, Hyun [21] used 1588 *E. coli* genomes from publicly available data to build SVM-based prediction models using amino acid sequence variants. The genomes used were of human origin and predominantly from the UK and USA. Hyun tested six antibiotics of which four overlapped with those in our study (AMC, CAZ, CIP and GEN). Compared to Hyun across these 4 antibiotics we achieved better AUC values in our study for 3 of 4 models (AMC: 0.83 vs 0.91, CAZ: 0.95 vs 0.97, GEN: 0.94 vs 0.95 [Hyun vs ours]), with our AUC for CIP slightly worse with a value of 0.97 compared to 0.98. Our sensitivity (recall) was better than in Hyun for all 4 antibiotics. Our models performed slightly worse against Hyun in three antibiotics for accuracy

(CAZ: 0.97 vs 0.92, CIP: 0.98 vs 0.94, GEN: 0.97 vs 0.90 [Hyun vs ours]) and precision (CAZ: 0.87 vs 0.88, CIP: 0.94 vs 0.96, GEN: 0.86 vs 0.94 [Hyun vs ours]). However, our AMC model performed better for both precision (0.67 vs 0.80) and accuracy (0.79 vs 0.85).

Similarly, in Moradigaravand [26] 1936 *E. coli* sequences (of which 1509 were also used in Ren, 2018) from human clinical and environmental sources were retrieved from collections in the UK, Pakistan, Syria, Sweden, the USA, and Belgium and used to predict AMR phenotypes using accessory gene presence-absence, matrix of the population structure, core genome SNPs, matrix of indels (insertions and deletions) and the year of isolation as features. Four machine learning classifiers were considered Random Forest, Gradient boosted decision trees, deep neural networks and logistic regression. Eleven antibiotics models were considered in Moradigaravand of which 6 overlapped with those of our study: AMC, AMP, CIP, CTX, CAZ (denoted CTZ in Moradigaravand) and GEN. Of these 6 antibiotics our dataset had better accuracy in 3 of 6 models (AMC: 0.81 vs 0.88, AMP: 0.93 vs 0.98, CIP 0.93 vs 0.94, CTX 0.97 vs 0.89, CAZ: 0.95 vs 0.93, and GEN: 0.97 vs 0.90 [Moradigaravand vs ours]). Our study had the same or better sensitivity (denoted R.RCL in (Moradigaravand et al. 2018)) in 5 of the 6 models (AMC: 0.64 vs 0.89, AMP: 0.96 vs 0.98, CIP 0.81 vs 0.97, CTX 0.92 vs 0.91, CAZ: 0.80 vs 0.80, and GEN: 0.81 vs 0.91 [Moradigaravand vs ours]). Comparing specificity our study achieved same or better specificity (denoted S.RCL in (Moradigaravand et al. 2018)) in 3 of 6 models (AMC: 0.60 vs 0.88, AMP: 0.96 vs 0.94, CIP 0.87 vs 0.88, CTX 0.93 vs 0.87, CAZ: 0.92 vs 0.96, and GEN: 0.98 vs 0.89 [Moradigaravand vs ours]).

Another *E. coli* study with a smaller sample size compared to ours was that of Her and Wu [20]. Her took 59 public database sequences (51 USA human isolates, 1 Taiwan human isolate, 1 Brazilian penguin isolate and 6 of unknown origin) and built machine learning predictors using radial SVM, Naïve Bayes, Adaboost, and Random Forest classifiers. Four different feature sets were used: core + accessory gene clusters, accessory gene clusters, CARD gene clusters and accessory gene cluster with CARD annotation. Of these the core + accessory gene cluster feature set is the most comparable with our k-mer features that cover both the core and accessory genome. Her and Wu studied 12 antibiotics of which eight overlapped with those studied in our work (AMP, AMS, CFZ, CAZ, FEP, GEN, CIP and SXT). Compared to Her and Wu [20] (looking at the most comparable feature set core + accessory gene clusters) our model gave the same or better AUC values for all the classifiers used in Her (AMP: 0.75 vs 1.00, AMS: 0.70 vs 0.97, CAZ: 0.84 vs 0.97, CFZ: 0.82 vs 1.00, FEP: 0.85 vs 0.94, GEN: 0.72 vs 0.95, CIP: 0.78 vs 0.97 and SXT 0.66 vs 0.99 [Her vs Ours]). Additionally, we achieved equal or better sensitivity values across 5 of 8 antibiotics compared Her and Wu (AMP: 0.5 vs 0.98, AMS: 0.53 vs 0.99, CAZ: 0.80 vs 0.80, CFZ: 0.86 vs 0.98, FEP: 0.90 vs 0.89, GEN: 1.00 vs 0.91, CIP: 1.00 vs 0.97 and SXT 0.49 vs 0.97 [Her vs Ours]), and better precision also in 5 antibiotics (AMP: 1.00 vs 0.99, AMS: 1.00 vs 0.95, CAZ: 0.81 vs 0.87, CFZ: 0.79 vs 0.99, FEP: 0.82 vs 0.89, GEN: 0.85 vs 0.86, CIP: 0.73 vs 0.94 and SXT 1.00 vs 0.98 [Her vs Ours]).

Interestingly, the gene ontology enrichment analysis indicated the presence of 7 genes (*prpD*, *mhpA*, *mhpB*, *mhpC*, *mhpF*, *mhpE*, *hcaB*) involved in the catabolism of aromatic compounds and in particular, phenylpropanoids [78], which suggests the possible involvement of these genes in antibiotic resistance mechanisms. A total of 73 metabolic and regulatory genes as well as those involved in phenylpropanoid catabolism were identified including *purM* and *purI* (purine metabolism), *argH* and *argA*, (arginine biosynthesis), *hisD*, *hisB* and *hisI* (histidine metabolism), *bioF* (biotin metabolism), *etk*, *narL and qseF* (two component regulatory systems). Purine and biotin metabolism have both been previously found to be in an enriched pathway correlated to antibiotic resistance [22].

Our machine learning analysis found a high number of known AMR-associated genes to be associated with the analysed AMR phenotypes. Among these genes, the bicyclomycin

resistance gene, *bcr*, was found to be significant in 11 of the antibiotic resistance models and identified as occurring in both IncFIA and IncFIB plasmids. This gene is part of the Bcr/CflA subfamily of multidrug efflux pumps, including the *floR* gene giving resistance to florfenicol antibiotics and BLAST search of our isolates against the CARD database [38] identified this gene as having a greater than 90% identity to the *floR* gene. This known AMR-associated gene appears to be widespread and a significant contributor to the AMR phenotypes seen in our cohort. The *floR* gene is increasingly prevalent in both animals [79] and humans [80] where it has been linked to transmission from chickens.

Notably, when we used the Jaccard/Tanimoto coefficient to search for correlation between resistance phenotypes and the AMR-associated genes, no significant results were obtained. Significant associations, as shown by the high prediction indicators, could only be obtained by using a more sophisticated approach such as machine learning. This shows the power of the learning approach for this type of analyses.

Gene sharing network analysis plays a fundamental role at providing detailed overviews of how AMR profiles and their associated most frequent sources are clustered for each antimicrobial, clearly indicating the elements of similarity and differentiations within and between communities. For example, our gene sharing networks showed that communities were often formed by drug-susceptible isolates from poultry and drug-resistant isolates of poultry and humans, highlighting the higher similarity among these three groups when compared to drug-susceptible *E. coli* isolates from humans. This result is in agreement with Johnson *et al.* (2007) [80], who showed comparable similarity results and suggested that drug-resistant human faecal *E. coli* isolates likely originate from poultry, whereas drug-resistant *E. coli* isolates from poultry likely originate from susceptible precursors in poultry.

In addition to the clustering and differentiation behaviours captured by indicators such as modularity, gene sharing networks allow also to identify sources acting as most likely bridges of transmission between host species and/or environments. To this purpose, the betweenness centrality indicator allows to isolate nodes (sources) with the highest number of connections, and thus acting as bridges between different communities, possibly underlying an important role in AMR gene transfer. For example, betweenness centrality indicates that for ampicillin/sulbactam the chickens are the hotspots for antibiotic resistance, while for the cefepime and kanamycin, resistance sharing involves multiple sources with no clear prevalence. These results suggest further investigation to uncover additional, biological supporting evidence.

We recognize that the differentiation of resistant and susceptible isolates seen in the clustermaps and networks is of limited value as they were based on the k-mers selected by a chi-squared test and extra tree classifier, during feature selection for the ML classifiers. Since the goal of the feature selection approach is to select the k-mers with the greatest separation between resistant and susceptible samples, the inputs into the cluster maps and networks predispose them to showing this separation. Nonetheless, the strength of this method is the ability to separate resistant samples into different communities uncovering possible patterns of antibiotic resistance sharing. For example, on Fig 3A, the ampicillin/sulbactam network indicates that the resistant samples are spread over 3 distinct communities (C0, C3 and C4); with the clustermaps it is possible to visualise why these communities are different. The majority of the susceptible samples are present on communities C1 and C2, while the network indicates a separation between these two communities, the clustermaps indicate that the samples are more similar and mixed between these two communities indicating the importance of using both methods together.

When investigating the drivers of the extensive ARG sharing among microbial communities in humans, animals, and the environment, we found the coexistence in the plasmid IncI2 of colistin resistance gene *mcr*-1 and gene *bla*$_{\text{CMX-M-97}}$. The *mcr*-1 gene has previously been

noted to give a fitness advantage to the IncI2 plasmid carrying it [63]. Though China is no longer using colistin as a growth promoter [63,81] the fitness advantage conferred by IncI2 plasmids supports potential transmission even in absence of antibiotic selection pressure.

Moreover, our findings showed a percentage of human isolates (29.3%) potentially sharing closely related AMR-associated mobile elements with those found in livestock. The percentage is not high, but still higher than what found (5%) in the previous UK study [37] and useful to identify potential transmission of resistance between hosts via mobile genetic elements. In particular, we mapped four possible hotspots for transmission of *E. coli*. In the farm environment, nose swabs from farmworkers showed similarity to isolates from both environmental and chicken samples. In the slaughterhouse environment, similar plasmids were present on carcasses, in water, and on the hand swabs of both the abattoir workers and their household members. The presence of clusters containing strains from chicken meat and humans is consistent with findings in the Netherlands [82]. The Netherlands during the years 2007 to 2009, like China, was one of the highest users of antimicrobial agents in food-production animals [83,84], resulting in high rates of drug resistance among these animals. Due to the decrease in the usage of antibiotics from 2010 onwards in the Netherlands, a decrease in *E. coli* resistance in poultry was also observed [84]. Likewise, differences observed between China and the UK study [37] could be caused by the UK being one of the lowest users of antimicrobial agents in food-production animals [83].

Most of the individual clusters obtained by gene sharing network analysis contained different STs, which is indicative of horizontal transfer of mobile genetic elements between lineages, as seen in previous work [37]. This potential horizontal transfer is supported by an *in vitro* study showing transfer of a $bla_{TEM-52}$ carrying plasmid from an avian *E. coli* strain to 2 human *E. coli* strain [85].

We acknowledge several limitations of our study. First, this study has been confined to a relatively small number of isolates and single colony picking. Although we covered two collection cycles, with so many potential features, ideally a larger number of isolates would allow refinement of the machine learning predictions. Other studies attempted the analysis of antimicrobial resistance on *E. coli* with machine learning and larger sample sizes [20,26,27,86]. However, the downside of requiring larger sample sizes is limitations in data availability, often requiring reliance on public databases and thus compromise on the type of available data and possible studies. For example, whilst not being able to rely on large amounts of data, we had the unprecedented possibility to perform a longitudinal study in the tightly controlled environments provided by using a farm and related slaughterhouse of the same commercial company.

Another potential limitation is that we only used short-read assemblies for the genetic analysis of plasmid and MGE transmission. Although suitable to identify phylogenetic correlations and hence potential transmission among stable plasmids, short-read assemblies offer lower resolution in tracking plasmids being mosaic and recombinant, compared to long-read sequencing [87–89]. Our approach may have limited the amount of plasmid diversity captured in this study. Also, our study lacks the wider contextualization possible only by using metagenomic approaches.

Our findings support the pressing need to extend surveillance on AMR both within and beyond clinical settings. In this work we found evidence of transmissible drug resistance in food-production animals and associated emergence of drug resistance in zoonotic pathogens. In several cases pathogens such as *E. coli* are part of the normal, human gut flora, thus the threat represented by drug resistance in these zoonotic pathogens is very high. LMICs lack of supportive regulation and widespread informal use of antibiotics might represent a risk of transmission for workers and their households. Further studies on the causes that foster or confine AMR exchange between environmental microbiota, human non-pathogenic and pathogenic bacteria, particularly during farming, are desirable.

## Methods

### Ethics statement

A key component of the programme of work was the recruitment of human faeces, hand swabs, nose swabs, chicken faeces and cloacal swabs and chicken carcasses for conventional microbiological analysis for presence of Escherichia coli.

Ethical approval has been obtained from the Research Ethic Committee in the School of Veterinary Medicine and Science at the University of Nottingham, and from Food Safety Risk Assessment, China National Center for Food Safety Risk Assessment, Beijing, China and assigned the following application IDs: 2340 180613 and 2018018.

### Sample collection and bacterial isolation

A total of 284 samples were taken over two independent 6-weeks broiler production cycles (March to June 2019) from a farm and connected slaughterhouse in Shandong province, China. The farm implements self-breeding uses the closed-end management model and contains on average 12000 chickens. A total of 284 isolates, were collected and included: soil surrounding the chicken barn (n = 12), chicken drinking water (n = 10), chicken feed (n = 16), farmworker faeces (n = 21), veterinary surgeon faeces (n = 3), farmworker hand swabs (n = 20), veterinary surgeon hand swabs (n = 3), worker nasal swabs (n = 20), veterinary surgeon nasal swabs (n = 3), chicken faeces (n = 60), chicken caecal droppings (n = 12), chicken cloacal swabs (n = 10), abattoir operator faeces (n = 11), abattoir operator nasal swabs (n = 11), abattoir operator hand swabs (n = 11), abattoir operator household faeces (n = 8), abattoir operator household nasal swabs (n = 8), abattoir operator household hand swabs (n = 9), abattoir waste water (n = 4), chicken carcass (n = 32). Connectivity of the samples is explained below for the *E. coli* positive samples. Farmed chickens were treated with kanamycin (aminoglycoside) for the first 5 days, tylosin (macrolide) from day 6 to 9, neomycin (aminoglycoside), from day 12 to 15, and florfenicol (chloramphenicol) from day 18 to 21 in the first production cycle and with amoxicillin (penicillin) from day 5 to 8 and neomycin (aminoglycoside) from day 9 to 12 in the second production cycle. To isolate *E. coli* strains an approximately 10g mixed fresh sample of 2–3 chicken faeces or an approximately 2g mixed fresh sample of 2–3 chicken caecal droppings was collected from the bottom of the chicken cage using a sterilized spoon. Chicken cloacal swab samples were collected using cotton-tipped swabs (108C.USE, Copan Diagnostics, Italy) from the same cage. Chicken carcass samples were collected in abattoir using sponge swabs (SS100NB, Hygiena International, Watford, UK). About 10g soil surrounding the chicken barn was collected at depth of 1-3cm within a distance of 5m from chicken barn. Not less than 20mL each of chicken drinking water of farm and waste effluent water from the slaughterhouse were collected from the water pipe or by using pipettes. About 10g each feed sample was collected using a sterilized spoon. Sterilized sampling spoons were used to collect 8g each of faeces from humans including chicken farm workers, the poultry veterinarian, slaughterhouse workers and slaughterhouse worker households. The hand swab sample and nasal swab samples were also collected using cotton-tipped swabs. All samples were collected using aseptic techniques, and then kept in secure containers at 4˚C during transportation to the laboratory and analysed within 24h.

### Bacteria isolation and diarrheagenic *E. coli* (DEC) identification

A quantity (volume) of 1 g (mL) sample of faeces, soil, feed, drinking water and polluted water each was vortexed with 9 mL of sterile buffered peptone water tube (BPW; Luqiao Inc., Beijing, China) for 1 min; nasal swab samples, hand swab samples and cloacal swab samples were vigorously vortexed with 9 mL BPW for 1 min in a test tube; chicken carcass sponge samples were homogenised with 10 mL BPW for 1 min in a stomacher bag. Approximately 1mL

dilution (of any of the above samples) was then added to 9 mL *E. coli* (EC) broth (Luqiao Inc.) and incubated at 37˚C for 16-20h in order to enumerate presumptive *E. coli* populations. A loopful of these solutions was then streaked onto an eosin-methylene blue (EMB) agar and MacConkey (MAC) Agar (Luqiao Inc.) and incubated at 37˚C for 18-24h. Typical *E. coli* colonies were screened and subsequently characterized by a Bruker MALDI Biotyper (Germany). The positive isolates identified were further confirmed by PCR using *E. coli*-specific primers ITS-F (5'-CAATTTTCGTGTCCCCTTCG-3') and ITS-R (5'- GTTAATGATAGTGTGTC-GAAAC-3'). Thermal amplification conditions were as follows: pre-incubation at 94˚C for 5 min, followed by 30 cycles of denaturation at 94˚C for 30s, annealing at 55˚C for 30s, elongation at 72˚C for 30s, and a final extension of 72˚C for 5 min. Out of 284 isolates analysed, 154 were identified as *E. coli* positive and included: chicken faeces (n = 49), farmworker faeces (n = 19), chicken carcasses (n = 15), cloacal (anal) swabs (n = 9), chicken caecal dropping (n = 9), farmworker hand swabs (n = 9), farmworkers nasal swabs (n = 8), abattoir operators faeces (n = 7) and their households' faeces (n = 7), animal feed (n = 6), abattoir polluted water (n = 3), abattoir operators' hand swabs (n = 3), soil surrounding the chicken barn (n = 3), abattoir operator households' nasal swabs (n = 2), abattoir operator households' hand swabs (n = 2), chicken drinking water (n = 2), abattoir operators' nasal swabs (n = 1). Chicken faeces samples were taken from small cages containing up to 10 chickens (cage ID numbers given in S1 Table), with up to two pooled samples taken per cage (each pooled sample containing 2–3 faeces). Sampling in this way gave a 5.2–11.4% probability of sampling the same chicken more than once at any time point (18.7–34.1% over both time points). A single pooled caecal sample containing 2–3 individual chicken samples gave rise to a 1.0–2.8% probability of sampling the same chicken more than once (5.2–11.4% over both time points). Cloacal swabs were taken from individual chickens and were all independent. Finally, humans were sampled at multiple body sites once per time point (human ID numbers given in S1 Table).

Confirmed *E. coli* strains were screened for five diarrheagenic pathotypes, EAEC (enteroaggregative *E. coli*; *aggR* and/or *astA* and/or pic), EPEC (enteropathogenic *E. coli*; escV), STEC (shiga toxin-producing *E. coli*; *stx1* and/or *stx2*), ETEC (enterotoxigenic *E. coli*; *lt* and/or *stp* and/or *sth*), and EIEC (enteroinvasive *E. coli*, i*nvE*), by the DEC multiplex PCR kit (Meizheng, Inc., Beijing, China).

## Antimicrobial susceptibility testing

Antimicrobial susceptibility to a panel of agents was determined by broth microdilution and interpreted according to the criteria based on the Clinical & Laboratory Standards Institute (CLSI) interpretive criteria (CLSI 2009). The minimum inhibitory concentrations (MIC) of 26 antimicrobial compounds were measured for the 154 *E. coli* isolates, and these included ampicillin (AMP), ampicillin/sulbactam (AMS), tetracycline (TET), chloramphenicol (CHL), trimethoprim/sulfamethoxazole (SXT), cefazolin (CFZ), cefotaxime (CTX), ceftazidime (CAZ), cefoxitin (CFX), gentamicin (GEN), imipenem (IMP), nalidixic acid (NAL), sulfisoxazole (SUL), ciprofloxacin (CIP), amoxycillin/clavulanic acid (AMC), polymyxin E (PE), polymyxin B (PB), minocycline (MIN), amikacin (AMK), aztreonam (AZM), cefepime (FEP), meropenem (MEM), levofloxacin (LEV), doxycycline (DOX), kanamycin (KAN) and streptomycin (STR). *E. coli* ATCC 25922 was used as a control for the antimicrobial susceptibility testing. The MIC50 value (the MIC required to inhibit 50% of cells), MIC90 value (the MIC required to inhibit 90% of cells), resistance rate and MIC distribution were calculated.

## DNA purification and extraction

The identified *E. coli* isolates were kept in brain heart infusion broth (BHI) medium with 20% glycerol at -80˚C and genomic DNA (gDNA) was purified using Omega EZNA Bacterial DNA

Kit (Omega Bio-tek, GA, USA). All the *E. coli* isolates were subjected to genomic DNA extraction in accordance with the manufacturer's protocol of E.Z.N.A. Bacterial DNA Kit (Omega Bio-Tek, Norcross, GA, USA).

### Library construction and whole genome sequencing

The template genomic DNA was fragmented by sonication to an insert size of 350 bp using NEBNext Ultra DNA Library Prep Kit for Illumina (NEB, USA) following the manufacturer's recommendations and index codes were added to attribute sequences to each sample and sequenced using an Illumina Hiseq 2500 PE150 achieving a median genome coverage of 270x (Range 140x-380x).

### Genome assembly and annotation

All sequences were pre-processed through readfq v10. To clean the data, reads containing low-quality bases (mean quality value $\leq$ 20) over 40% were removed. Reads with greater than 10% unidentified bases (N) were removed as well as the adapters. The whole-genome shotgun sequencing produced 154 high-quality reconstructed *E. coli* genomes with a N50 larger than 50,000 and less than 250 contigs (S1 Table). Cleaned data were processed for genome assembly with SPAdes v3.13, and QUAST v4.5 was used for assessing the assembly. The contigs with length shorter than 500 nucleotides were filtered out. The completeness and contamination of genomes were assessed through checkM with the lineage_wf pipeline. Genomes were annotated with Prokka v1.14.5 [90] using default parameters with—addgenes–usegenus.

### Screening of annotated genes against ABR databases

The whole genome sequences were screened against the CARD [38] database with a minimum coverage and identity of 90% to identify known AMR-associated genes in the isolate cohort. In addition the annotated genes obtained from Prokka v1.14.5 [90] were screened against the CARD [38], ARG-Annot [40] and Resfinder [39] databases using Abricate [91] and the NCBI AMRfinder [41] database; all the comparative analyses have been done with a minimum coverage and identity of 90%. For some genes identified in the AMR databases, the gene name differs from the name assigned by Prokka, due to synonymous genes. In these cases, we have opted to use the more widely recognised gene name as reported in the AMR databases throughout the text, though the original name can be found in S3 Table. Additionally, gene names were screened against the MutationDB database that records genes reported in literature to have undergone mutations due to antibiotic stress [49]. Where available, gene accession numbers were added to S3 Table.

### In silico subtyping identification and standard index of association ($I_A$) calculation

Sequence types were identified through MLST which mapped the sequences to the PubMLST *E. coli* MLST database. Clonal complexes were annotated from known CC types in the MLST database. Phylogroups were identified using in silico Clermont typing [92]. Serotypes were identified through the EcOH database [92] using Abricate [91].

   Linkage disequilibrium of the 154 isolates in our cohort was evaluated using the standardized index of association ($I^S_A$) [43], which estimates the homologous recombination for the cohort by assessing the linkage disequilibrium among the seven MLST loci. The LIAN Ver. 3.7 program was used to calculate the $I^S_A$ for all the isolates and for a subset of them (one isolate for each ST type) from the ratio of the variance of observed mismatches in the test set ($V_D$) to

the variance expected for a state of linkage equilibrium ($V_e$), scaled by the number of loci used in the analysis (L) [43,93].

$$I_A^S = \frac{1}{L-1}\left(\frac{V_D}{V_e} - 1\right)$$

The significance of $I_A^S$ was determined by a Monte Carlo simulation with $10^5$ resamplings.

## Whole-genome phylogenetic analysis and network analysis based on pairwise SNP alignment

All annotated genomes were taken as input for pan-genome analysis with core gene alignments through Roary v3.13 [94]. IQTree v2.0.3 [95] was then used to construct the phylogenetic trees from the core genome alignment. Fifty-three different nucleotide replacement models were tested automatically with the GTR (+F+R5) replacement model selected as the best model. The Ultrafast bootstrap algorithm was used with 1000 replicates to assess branch support. The phylogenetic trees were subsequently visualised through iTOLv5 [96]. The core genome alignment was taken as input to produce a file of core gene SNPs present in the cohort using snp-sites [97]. A network of *E. coli* isolates collected from different sources in the farm (human, chicken and environmental) was created using a pairwise hamming distance comparison based on SNPs in the core genome. Each node represents an isolate while the edge represents the hamming distance between two isolates multiplied by the total number of SNPs found in our cohort (133,631 SNPs). A threshold of 15 or less SNPs difference was used to filter the edges in the network as suggested by Ludden *et al* (2019) [37].

## Machine learning analysis

Machine learning methods were used to search for the features in the genome sequence of each isolate which could strongly correlate to resistance to each one of the of the 26 selected antimicrobials. Sample genomes were first split into overlapping 13-mers using GenomeTester [98] to produce a feature table for all samples. The AMR phenotype (resistant, susceptible or intermediate) of each sample (S1 Table) was used as the class label with intermediate phenotypes neglected. As the classes were unbalanced a synthetic minority oversampling technique [99] was used during the training phase of the classifiers to balance the proportion of classes in the data set. The Python package Scikit-learn [100] was used to make the classification and to select the most important features.

Fig 2A indicates the pipeline with the five steps used in this work. The Python package Scikit-learn [100] was used to make the classification and to select the most important features.

1. A two-step approach to overcome the disadvantages of applying a chi-square test in multiple comparisons is used: i) all the k-mers that have a p-value > 0.05 (the null hypothesis was not rejected) were discarded; ii) the ones with a p-value < 0.05 were used as input to an Extra Tree Classifier (randomized decision trees) and the k-mers with a Gini feature importance above the overall mean were selected.

2. A panel of machine learning methods (logistic regression (LR), linear support vector machine (L-SVM), radial basis function support vector machine (RBF-SVM), extra tree classifier, random forest, adaboost, xgboost, naïve bayes, linear discriminant analysis and quadratic discriminant analysis) were then run using as input the k-mers uncovered on the first step and their performances were evaluated.

3. A BLASTN approach on the whole genome sequence of each sample to identify the genes related to each k-mer selected on the first step.

4. The genes with a relative presence higher than 30% between resistant and susceptible samples are selected to be input for the clustermaps and gene sharing networks.

5. The genes selected in the previous step are evaluated in AMR public databases (CARD [38], ARG-Annot [40], ResFinder [39]) to confirm if they are known AMR genes.

Nested Cross-validation (NCV) [101] was employed to assess the performance and select the hyper-parameters of the proposed classifiers. In NCV, there was an outer loop split of the data set into test and training sets. For each training set, a grid search (inner loop) was run, to find the best hyper-parameters of the classifier using accuracy as a performance metric. Then, the test set was used to score the best classifier found in the inner loop. Thirty iterations were carried out, wherein each iteration an NCV was employed. The inner loop of the NCV found the best hyper-parameters of each classifier using stratified 3-fold cross-validation; the outer loop measured the ROC-AUC (receiver operating characteristic area under the curve) accuracy, sensitivity, specificity and Cohen's kappa of the test data set (unseen in the inner loop for the training) using 5-fold stratified cross-validation, to compare all the classifiers [102]. A synthetic minority oversampling technique (SMOTE) was used to reduce the impact of unbalanced classes in the antimicrobial label groups. The number of splits in the nested cross validation and the number of k-nearest neighbours for SMOTE necessitated at least 12 samples in each class. The prediction metrics accuracy (TP +TN/(P+N)), sensitivity (true positive rate: TP/P), specificity (true negative rate: TN/N), Area Under the ROC Curve (AUC) and Cohen's Kappa were used, as typically used to measure ML performance, see [22,30,103–105]. Violin plots from the Seaborn package [106] were used to show the final prediction metrics.

To compare the results obtained by the 10 different classifiers used, a Friedman Statistical F-test was employed. The Friedman test $F_F$ with Iman-Davenport correction [107] is employed for statistical comparison of multiple classifiers as suggested by [108]. First, we rank the algorithms for each dataset separately, i.e., the best algorithm gets ranking 1, the second best ranking 2, and so on. In case of ties, average ranks are assigned. Next, we apply the $F_F$ and verify if the null hypothesis is rejected. After, the post-hoc Nemenyi test [108] was used to find if there is a single classifier or a group of classifiers that differ in terms of their average rank after the $F_F$ test has rejected the null hypothesis that the performance of the comparisons on the groups of data is similar.

To analyse if the antibiotics studied had enough samples to make the test representative, a wrapper backward selection (WBS) approach in terms of samples was used, i.e. if initially we have 100 samples we are first going to test the model using all 99 possible sample combinations (leaving one out), the worst performance is identified and the sample that was left out on this performance is removed for the next iteration. This is done until the minimum number of samples (12) is reached for one of the classes. This minimum number of samples is a requirement for the SMOTE approach used in the classification framework.

## Annotation and abundance of the ARGs in the farm and slaughterhouse

Where the machine learning was able to predict the antimicrobial class based on k-mers, these were then used to search the genome for genes that contained the k-mers. The k-mers were mapped to the pangenome of the 154 isolates using a BLASTN [109] query with the parameters: e-value = 1000, word-size = 13, gap-open = 0, gap-extend = 0, out-fmt = 5. Genes with an identity of >70% and coverage of >70% were considered to be variants of the same gene and hence were discounted as duplicates, as done in previous literature [110], however a more

stringent threshold was used to ensure all gene variants were accounted for. The k-mer hit count (how many k-mers mapped to each identified gene) of the genes identified was then assessed for statistical significance at a significance level of 0.05 using a binomial exact test, with the probability of a gene hit based on the length of the gene and number of k-mer combinations possible per gene. To analyse the co-presence of the *aadA2* gene, a BLASTN search was used to find contigs in the WGS isolates with the *aadA2* sequence as the query and a 95% identity threshold. Genes present in contigs matching to the *aadA2* sequence were selected from the annotated whole genome sequences and summarised by isolate.

To select the most relevant genes we analysed their presence in the resistant (Pr) and susceptible (Ps) isolates. Where a gene was considered as present in a sample if at least one k-mer recognised as discriminant for the specific AMR profile by the classifiers could be mapped to that specific gene. A gene was considered not present in a given sample if all the discriminant k-mers associated to that gene were absent in that specific isolate. If the difference |Pr–Ps| was higher than δ the gene was selected and identified as important to differentiate resistant and susceptible isolates; in our case δ was equal to 30%. To analyse the co-occurrence of the genes that considered to be the most relevant the Jaccard coefficient was calculated in a pairwise manner between the two vectors of gene presence/absence (binary data) in each assembled contig within the cohort.

## Functional enrichment analysis

The functional characterisation of the genes not annotated in AMR public databases that presented a |Pr-Ps| > 30% was done using StringDB v11.5 [59], that scans against both the Gene Ontology (GO) [60] terms (biological process and molecular function) and Kyoto Encyclopaedia of Genes and Genomes (KEGG) pathways [111]. All strains of the Escherichia genus present in StringDB were queried and *E coli* K12 MG655 gave the best match with 211 out of 278 matched genes (75.9%) while *E coli* O157:H7 gave a match to 200 out of 278 genes (71.9%). The functional enrichment analysis was run on both *E. coli* strains with default settings. The hypergeometric test with false discovery rate correction were used to analyse statistically the GO and KEGG pathways analysis. To visualize the results, a chord diagram was produced using the library nxviz v0.6.3 in python.

## AMR gene sharing network analysis for the chicken farm and slaughterhouse environments

To identify the hot-spots of resistant bacteria and AMR-conferring genes and investigate their transmissibility across humans, animals and their environment, within and between farm and slaughterhouse environments a novel graph network was constructed based on the work of Bernard *et al.* (2016) [112]. The AMR gene sharing network for each one of the 21 selected antimicrobials was built using the ARGs recognised as discriminant by the machine learning approach and fulfilling the |Pr–Ps|>30% selection criteria. To data mine the graph network, a community detection algorithm was applied using the NetworkX [113] and the community python-louvain [114] libraries. A community is a set of nodes (cluster) that are densely connected. It is typical to identify social behaviours [115]. In order to find the communities in a graph, the algorithm partitions the nodes that maximise the modularity using the Louvain heuristics [116]. In the gene sharing network, each node represents a sample (colour and shape indicates the metadata associated with each sample), and an edge connecting two nodes represents the Euclidean distance between two samples based on the presence/absence of the discriminant k-mers related to the genes. The networks were analysed to assess modularity and betweenness centrality. Modularity measures the strength of division of a network into

communities. It is usually expressed in the 0–1 range, and higher modularity values indicate a network with tightly connected samples within the communities, but sparse connections between communities. Betweenness centrality is computed for all the nodes in the network, and -for each node- quantifies how many paths connecting two other nodes pass through that node.

## Plasmid reconstruction and hierarchical clustering

Plasmids were reconstructed using the MOB-recon algorithm in the MOB-suite package [61], using default settings. The presence/absence of each plasmid in the cohort was visualised in iTOLv5. Reconstructed plasmid sequences were searched using BLASTN to identify which genes from the WGS pangenome were plasmid located (search with the pangenome as a query with a 95% identity threshold, default settings). The 1113 genetic determinants of resistance were then checked to see if they were present on any plasmid in the isolate and assumed to be chromosome based if not present. Each reconstructed plasmid was annotated using Prokka v1.4.5 [90]. For each identified replicon type with more than 5 reconstructed plasmid sequences, gene presence/absence was analysed using Roary v3.13 [94]. The gene presence/absence output from Roary was used as input for hierarchical clustering of the reconstructed plasmids in R [117]. A binary distance matrix (scaled between 0 and 1) of the reconstructed plasmids was created then the plasmids were clustered hierarchically using the *hclust* function in R (with method = 'Jaccard'). The heights of the clusters represent the Jaccard distance between plasmids.

## Supporting information

**S1 Fig. Single nucleotide polymorphism network analysis of highly connected isolates.** Network diagram showing pairwise connections between human, chicken and environmental isolates with less than 15 pairwise SNP differences. The lines between pairs of isolates are colour-coded by SNP number.
(PDF)

**S2 Fig. Nemenyi Diagram.** Nemenyi post-hoc test for the performance metrics (A) AUC, (B) accuracy, (C) sensitivity, (D) specificity, (E) precision and (F) Cohen's Kappa score for the following classifiers: logistic regression, linear SVM, RBF-SVM, extra tree classifier, random forest, adaboost, xgboost, naïve bayes, linear discriminant analysis (LDA) and quadratic discriminant analysis (QDA).
(PDF)

**S3 Fig. Learning Curves.** Learning curves of the 21 antibiotics using a wrapper backward selection to evaluate the training and testing performance whilst decreasing the number of samples. (A) The original number of samples for each antibiotic and 5 nested cross validation iterations with an RBF-SVM as the main classifier; (B) SMOTE approach as a pre-processing step to increase the number of samples by adding synthetic samples to the minority class for each antibiotic and 5 nested cross validation iterations with an RBF-SVM as the main classifier. In both cases, the same features used to acquire the performances in the Fig 2 and S2 Table are used and kept for all the iterations of the WBS. The red lines indicate the training performance, the green lines the testing performance and the blue vertical line the original number of samples.
(PDF)

**S4 Fig. Networks of AMR genes shared across host species (animals and workers) and environments (farm and slaughterhouse) for 18 antimicrobials.** The resistome networks for

amoxycillin/clavulanic acid, ampicillin, aztreonam, ceftazidime, cefoxitin, chloramphenicol, ciprofloxacin, cefotaxime, doxycycline, cefazolin, gentamycin, levofloxacin, minocycline, nalidixic acid, streptomycin, sulfisoxazole, trimethoprim/sulfamethoxazole and tetracycline from (A) to (R). For each network: (i) Panel on the left, indicates the communities and their respective numbers found using Louvain heuristics. Each community is indicated with a distinct colour. Each community is a set of nodes or clusters, that are densely and connected with statistical significance and identically coloured. For each network the node represents a sample and is shown in a distinct colour and shape depending on the metadata of the sample (e.g., source, AMR profile, location). In all panels the nodes are separated according to the Euclidean distance between the isolates; The central panel shows the AMR phenotype (resistant or susceptible) as the shape of the node and the source of the samples (human, chicken or environmental) as the colour of the node. An edge represents the Euclidean distance between two samples, and it is coloured according to the associated statistical value. The panel on the right indicates the location of the sample (farm or slaughterhouse) as the shape of the node and the type of sample as the colour of the node. (ii) Clustermap showing the genes associated with the AR discriminant k-mers used to build the networks. The columns on the left show the metadata (Class, Location, Source, Type, Clonal Complex, Phylogroup, Community number). The presence of the genes (in terms of its related k-mers) is indicated in black, while the absence is indicated in white).
(PDF)

**S5 Fig. Machine learning hits to transcription factor binding sites in intergenic regions of the whole genome sequences.** The heatmap shows for each antibiotic model, transcription factors where the k-mers selected by machine learning mapped to binding sites for the transcription factor and the proportion of individual isolates hit were more than 30% different between the resistant and susceptible isolates. The presence of a binding site hit is indicated in pink, with blue denoting no hit. Antibiotic models with no significant hits were neglected from the figure.
(PDF)

**S6 Fig. Functional enrichment analysis of genes identified by machine learning.** (A) Circos plot of KEGG pathways found to be significantly enriched in the 278 unique genes found by the machine learning compared to the whole genome background not annotated as ARM in public databases. (B) Circos plot of gene ontology molecular functions found to be significantly enriched in the 278 unique genes found by the machine learning compared to the whole genome background not annotated as AMR in public databases.
(PDF)

**S7 Fig. Proportion of genes located on plasmid and hierarchical clustering of isolates based on similarity of plasmid type.** (A) Boxplot showing the proportion of genes located on plasmids per isolate grouped by host type. Isolates from chicken were found to have significantly more genes located on plasmids that human farmworkers and abattoir worker household members. (B) Hierarchical clustering of plasmids with low number of significant genes grouped by replicon type. Clustering was based on gene presence or absence. Green underscores indicate areas of high similarity (>0.996 WGS ANI value) between isolates from different hosts, suggestive of transmission between hosts.
(PDF)

**S1 Table. Characteristics of the single isolates, including collection details, typing data, genomic features, and results of antibiotic susceptibility testing, CARD ARGs and ST**

**types.**
(XLSX)

**S2 Table. Performance metrics (AUC, Accuracy, Sensitivity, Specificity, and Cohen's Kappa value) results for the RBF-SVM classification into resistance or susceptible for 21 antibiotics and for the results of the other 9 classifiers used (logistic regression, linear SVM, extra tree classifier, random forest, adaboost, xgboost, naïve bayes, linear discriminat analysis and quadratic discriminant analysis); Genes in which k-mers selected as features for the machine learning were found; Jaccard coefficients indicating the gene co-occurrence for all genes found to be significantly correlated with AMR phenotypes.**
(XLSX)

**S3 Table. List of the 361 genes found to be significantly correlated with AMR phenotypes by machine learning.** For each antibiotic, a sheet showing the frequencies of each gene in each sample are shown.
(XLSX)

**S4 Table. Gene ontology enrichment analysis (molecular function and biological process) and KEGG pathways results for the 278 unique genes that were not annotated as AMR in public databases and were selected due to their significance when comparing resistant and susceptible samples (|Pr-Ps| > 30%).**
(XLSX)

**S5 Table. List of the 361 genes found to be significantly correlated with AMR phenotypes, as found by supervised machine learning and their associated plasmids.** Known AMR and virulence genes are noted. Presence in a plasmid is indicated in pink and absence in blue.
(XLSX)

**S6 Table. Presence/absence of all annotated genes located on plasmids of different replicon types.** Presence in a plasmid is indicated by 1 and absence by 0.
(XLSX)

## Acknowledgments

We thank the farm and slaughterhouse workers and their households for their generosity and trust, without which this study would not have been possible. We wish to thank the Future Food Beacon at the University of Nottingham for support and Richard Emes for fruitful discussions and for reading and commenting on the manuscript and Ali Rohan for helping with the response to the reviewers.

## Author Contributions

**Conceptualization:** Zixin Peng, Alexandre Maciel-Guerra, Michelle Baker, Xibin Zhang, Paul Barrow, Dov Stekel, Longhai Liu, Junshi Chen, Fengqin Li, Tania Dottorini.

**Data curation:** Alexandre Maciel-Guerra, Michelle Baker.

**Formal analysis:** Alexandre Maciel-Guerra, Michelle Baker, Yue Hu, Ning Xue.

**Funding acquisition:** Zixin Peng, David Renney, Longhai Liu, Tania Dottorini.

**Investigation:** Zixin Peng, Xibin Zhang, Wei Wang, Jia Rong, Jing Zhang.

**Methodology:** Zixin Peng, Xibin Zhang, Longhai Liu, Fengqin Li, Tania Dottorini.

**Supervision:** Junshi Chen, Fengqin Li, Tania Dottorini.

**Visualization:** Alexandre Maciel-Guerra, Michelle Baker, Yue Hu, Ning Xue.

**Writing – original draft:** Alexandre Maciel-Guerra, Michelle Baker, Tania Dottorini.

**Writing – review & editing:** Alexandre Maciel-Guerra, Michelle Baker, Paul Barrow, Paul Williams, Tania Dottorini.

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
