## [Decision Letter · Decision Letter 0]

17 Jan 2022

Dear Dr Dottorini,

Thank you very much for submitting your manuscript "Whole genome sequencing and gene sharing network analysis powered by machine learning identifies antibiotic resistance sharing between animals, humans and environment in livestock farming" for consideration at PLOS Computational Biology.

As with all papers reviewed by the journal, your manuscript was reviewed by members of the editorial board and by several independent reviewers. In light of the reviews (below this email), we would like to invite the resubmission of a significantly-revised version that takes into account the reviewers' comments.

The reviewers have raised significant concerns about the method and scope, which need to be addressed before the manuscript can be considered for publication.

Primary concerns expressed were that:

- The novelty and choice of the methods have not been adequately justified.

- The machine learning pipeline needs further details.

- The study needs a better contextualisation to highlight the novelty.

- The sample size is fairly small and therefore the findings may not be generalisable to other settings. The small size may also affect the performance of the models.

- The raw data and annotated assemblies need to be shared in public repositories, with accession numbers provided in the manuscript.

We cannot make any decision about publication until we have seen the revised manuscript and your response to the reviewers' comments. Your revised manuscript is also likely to be sent to reviewers for further evaluation.

Sincerely,

Danesh Moradigaravand

Guest Editor

PLOS Computational Biology

Ville Mustonen

Deputy Editor

PLOS Computational Biology

Reviewer's Responses to Questions

**Comments to the Authors:**

Reviewer #1: In this study, the authors proposed a machine learning model to identify antibiotic resistance sharing between animals, humans and environment in livestock farming. Whole genome sequencing and gene sharing network analysis are used. The authors reached a promising performance, however, some major points should be addressed:

1. More literature review should be added to show more related works to this study.

2. The authors used logistic regression, linear support vector machine and radial basis function kernel support vector machine (RBF-SVM) as their algorithms. Why was extra tree used in Fig. 2?

3. The use of inconsistent cross-validation method, i.e., sometimes using nested CV, sometimes using 5-fold or 3-fold.

4. This study requires an external validation data to evaluate the performance of models on unseen data.

5. The authors should compare the predictive performance to previously published works on the same problem/data.

6. Measurement metrics (i.e., AUC, Accuracy, Sensitivity, Specificity, ...) have been used in previous bioinformatics studies such as PMID: 34915158, PMID: 34812044. Thus, the authors are suggested to refer to more works in this description to attract a broader readership.

7. "..." parts in Fig. 2 should be filled completely.

8. Why did the authors only test 3 machine learning models? Other advanced algorithms should be assessed also.

Reviewer #2: The overall idea of the study is good, and the objectives are clear. However, I have some significant concerns.

1) The study size is very small. It was tough for me to aggregate all the relevant information. Please add a table (not as a supplement) where you put the number of samples for each drug, i.e., the counts of resistant and susceptible samples per drug.

2) The machine learning approach is very similar to the recently published paper by Ren et al. (pubmed:34613360). However, they used fewer antibiotics but far more samples. Please discuss your findings in the context of this study and maybe check whether you found similar genetic information / SNPs.

3) I doubt the statistical robustness of non-linear kernels here. The sample size is tiny (still unclear, see 1)), with a lot of features (here: SNPs) and many different antibiotics. These things make it very likely that the results are not reproducible.

**Have the authors made all data and (if applicable) computational code underlying the findings in their manuscript fully available?**

Reviewer #1: **No: **Only code is shared, no data.

Reviewer #2: Yes

PLOS authors have the option to publish the peer review history of their article (what does this mean?). If published, this will include your full peer review and any attached files.

Reviewer #1: **Yes: **Nguyen Quoc Khanh Le

Reviewer #2: No
---

## [Decision Letter · Decision Letter 1]

14 Mar 2022

Dear Dr Dottorini,

We are pleased to inform you that your manuscript 'Whole genome sequencing and gene sharing network analysis powered by machine learning identifies antibiotic resistance sharing between animals, humans and environment in livestock farming' has been provisionally accepted for publication in PLOS Computational Biology.

Best regards,

Danesh Moradigaravand

Guest Editor

PLOS Computational Biology

Ville Mustonen

Deputy Editor

PLOS Computational Biology

Reviewer's Responses to Questions

**Comments to the Authors:**

Reviewer #1: My previous comments have been addressed.

Reviewer #2: The authors addressed my concern adequately.

**Have the authors made all data and (if applicable) computational code underlying the findings in their manuscript fully available?**

Reviewer #1: None

Reviewer #2: Yes

PLOS authors have the option to publish the peer review history of their article (what does this mean?). If published, this will include your full peer review and any attached files.

Reviewer #1: **Yes: **Nguyen Quoc Khanh Le

Reviewer #2: No

---

## [Editor Report · Acceptance letter]

22 Mar 2022

PCOMPBIOL-D-21-02089R1 

Whole genome sequencing and gene sharing network analysis powered by machine learning identifies antibiotic resistance sharing between animals, humans and environment in livestock farming

Dear Dr Dottorini,

I am pleased to inform you that your manuscript has been formally accepted for publication in PLOS Computational Biology. Your manuscript is now with our production department and you will be notified of the publication date in due course.

With kind regards,

Agnes Pap
